

# Heterogeneous N₂O₅ uptake coefficient and production yield of ClNO₂ in polluted northern China: Roles of aerosol water content and chemical composition

Yee Jun Tham[1,2], Zhe Wang[1], Qinyi Li[1], Weihao Wang[1], Xinfeng Wang[3], Keding Lu[4], Nan Ma[5], Chao Yan[2], Simonas Kecorius[5], Alfred Wiedensohler[5], Yuanhang Zhang[4], and Tao Wang[1]

[1]Department of Civil and Environmental Engineering, The Hong Kong Polytechnic University, Hong Kong, China
[2]Institute for Atmospheric and Earth System Research/Physics, University of Helsinki, 00014, Helsinki, Finland
[3]Environment Research Institute, Shandong University, Jinan, Shandong, China
[4]State Key Joint Laboratory of Environmental Simulation and Pollution Control, College of Environmental Sciences and Engineering, Peking University, Beijing, China
[5]Leibniz Institute for Tropospheric Research, Permoserstr. 15, 04318 Leipzig, Germany

*Correspondence to: Z. Wang (z.wang@polyu.edu.hk) and T. Wang (cetwang@polyu.edu.hk)*

**Abstract.** Heterogeneous uptake of dinitrogen pentoxide (N₂O₅) and production of nitryl chloride (ClNO₂) are important nocturnal atmospheric processes that have significant implications for the production of secondary pollutants. However, the understanding of N₂O₅ uptake processes and ClNO₂ production remains limited, especially in China. This study presents a field investigation of the N₂O₅ heterogeneous uptake coefficient ($\gamma$(N₂O₅)) and ClNO₂ production yield ($\phi$) in a polluted area of northern China during the summer of 2014. The N₂O₅ uptake coefficient and ClNO₂ yield were estimated in 10 selected cases using simultaneously measured concentrations of ClNO₂ and particulate nitrate. The determined $\gamma$(N₂O₅) and $\phi$ values varied greatly, with an average of 0.022 for $\gamma$(N₂O₅) (in range of 0.006–0.034) and 0.34 for $\phi$ (range, 0.07–1.04). The variations in $\gamma$(N₂O₅) could not be fully explained by the previously derived parameterizations of N₂O₅ uptake that consider nitrate, chloride, and the organic coating. Heterogeneous uptake of N₂O₅ was found to have a strong positive dependence on the relative humidity and aerosol water content. This result suggests that the heterogeneous uptake of N₂O₅ in Wangdu is governed mainly by the amount of water in the aerosol, a phenomenon that differs from other field observations in the United States and Europe. Laboratory-derived parameterization also overestimated the ClNO₂ yield. The observation-derived $\phi$ showed a decreasing trend with an increasing ratio of acetonitrile to carbon monoxide, an indicator of biomass burning emissions, which suggests a possible suppressive effect on the production yield of ClNO₂ in the plumes influenced by biomass burning in this region. The findings of this study illustrate the need to improve our understanding and to parameterize the key factors for $\gamma$(N₂O₅) and $\phi$ to accurately assess the photochemical and haze pollution.



# 1 Introduction

The nocturnal heterogeneous reaction of dinitrogen pentoxide ($N_2O_5$) with aerosols is a loss pathway of $NO_x$ and a source of aerosol nitrate and gas-phase nitryl chloride ($ClNO_2$) (Brown et al., 2006; Osthoff et al., 2008; Thornton et al., 2010; Sarwar

5   et al., 2014) and thereby has important implications on air quality (e.g., Li et al., 2016). The process begins with the accumulation of gas-phase nitrate radical ($NO_3$) after sunset via the oxidation of nitrogen dioxide ($NO_2$) by $O_3$ and further reaction of $NO_3$ with another $NO_2$, yielding a $N_2O_5$. The accommodation of $N_2O_5$ on the aqueous surface of the aerosol (R1) and reaction with liquid water ($H_2O$) leads to the formation of a protonated nitric acid intermediate ($H_2ONO_2^+$) and a nitrate ($NO_3^-$) (R2; Thornton and Abbatt, 2005; Bertram and Thornton, 2009).

$$N_2O_5 \text{ (g)} \leftrightharpoons N_2O_5 \text{ (aq)} \tag{R1}$$
$$N_2O_5 \text{ (aq)} + H_2O \text{ (l)} \leftrightharpoons NO_3^-\text{(aq)} + H_2ONO_2^+\text{(aq)} \tag{R2}$$

The $H_2ONO_2^+$ will proceed by reacting with another $H_2O$ to form an aqueous nitric acid ($HNO_3$; R3). If chloride ($Cl^-$) is present in the aerosols, the $H_2ONO_2^+$ will undergo another pathway to produce a nitryl chloride ($ClNO_2$) through R4, which is a dominant source of highly reactive chlorine radicals in the troposphere (e.g., Riedel et al., 2012; 2014).

$$H_2ONO_2^+\text{(aq)} + H_2O \text{ (l)} \rightarrow H_3O^+\text{(aq)} + HNO_3\text{(aq)} \tag{R3}$$
$$H_2ONO_2^-\text{(aq)} + Cl^-\text{(aq)} \rightarrow ClNO_2 + H_2O\text{(l)} \tag{R4}$$

The heterogeneous loss rate of $N_2O_5$ ($k(N_2O_5)_{het}$) and the $ClNO_2$ production rate ($p(ClNO_2)$) are fundamentally governed by the probability of $N_2O_5$ lost upon collision with particle surface area in a volume of air (i.e., uptake coefficient,

20   $\gamma(N_2O_5)$) and the $ClNO_2$ yield ($\phi$), which is defined as the branching ratio between the formation of $HNO_3$ via R3 and $ClNO_2$ via R4. Assuming that the gas-phase diffusion to the aerosol surfaces is negligible, their relationship can be described by equations (1) and (2), in which $c_{N2O5}$ is the average molecular speed of $N_2O_5$ and $S_a$ is the aerosol surface area.

$$k(N_2O_5)_{het} = \frac{1}{4} c_{N2O5} \gamma(N_2O_5) S_a \tag{Eq 1}$$
$$p(ClNO_2) = k(N_2O_5)_{het}[N_2O_5]\phi \tag{Eq 2}$$

$N_2O_5$ uptake has been shown in the laboratory to be susceptible to changes in the water content, chloride, nitrate, and organic particle coatings in aerosols (e.g., Mentel et al., 1999; Bertram and Thornton, 2009). The presence of liquid water on the aerosols allows the accommodation of $N_2O_5$ (R1) and acts as a medium for the solvation process of $N_2O_5$ (R2). It has been found that $N_2O_5$ uptake is significantly enhanced in humid conditions than in dry conditions (e.g., Hallquist et al., 2003;

30   Bertram and Thornton, 2009; Gržinic et al., 2015). Higher loading of $NO_3^-$ in the aerosol can dramatically decrease $N_2O_5$ uptake by reversing the solvation/ionization process of $N_2O_5$, shifting the equilibrium in R2 to the left to reproduce $N_2O_5$, which can be diffused out of the aerosol (known as the "nitrate suppression" effect). The rate of reversible reaction of R2 (i.e., $H_2ONO_2^+$ with $NO_3^-$) was documented to be 30 to 40 times faster than the reaction of $H_2ONO_2^+$ with liquid water in R3



(Bertram and Thornton, 2009; Griffiths et al., 2009). The presence of Cl⁻ in the aerosol, in contrast, can enhance the reactive uptake because Cl⁻ reacts effectively with $H_2ONO_2^+$ (in R4), thus negating the "nitrate suppression" effect by shifting the equilibrium in R2 to the right (Finlayson-Pitts et al., 1989; Bertram and Thornton, 2009). The uptake of $N_2O_5$ can also be hindered by the presence of organics, because the organic coating layer on the aerosol could lower the liquid water content

and/or limit the surface activity, thus suppressing the accommodation of $N_2O_5$ (e.g., Cosman et al., 2008; Gaston et al., 2014).

As for the $ClNO_2$ yield from $N_2O_5$ heterogeneous reactions, it was found to be dependent on the fate of $H_2ONO_2^+$ and thus on the relative amount of Cl⁻ and water content (Benhke et al., 1997; Roberts et al., 2009; Bertram and Thornton, 2009).

Therefore, φ can be expressed by the following equation (Eq.3).

$$\phi_{param.} = \frac{1}{\frac{k_{R3}[H_2O]}{k_{R4}[Cl^-]} + 1} \qquad (Eq\ 3)$$

Roberts et al. (2009) reported that the coefficient rate of $k_{R4}$ is about 450 times faster than that of $k_{R3}$, indicating that $H_2ONO_2^+$ proceeds more favorably via R4, even with a small amount of Cl⁻. However, some laboratory experiments have suggested that the presence of halides (i.e., bromide), phenols, and humic acid may significantly reduce the φ (e.g.,

Schweitzer et al., 1998; Ryder et al., 2015).

The parameterization of γ($N_2O_5$) and φ as a function of the aerosol water content and aerosol chemical composition, derived based on the findings of the laboratory studies mentioned above (e.g., Antilla et al., 2006; Bertram and Thornton, 2009; Davis et al., 2008), has recently been compared with the ambient observations in different environments (Phillips et al.,

2016; Chang et al., 2016; Wang Z. et al., 2017). Large discrepancies were observed between the γ($N_2O_5$) and φ values determined in the fields and the laboratory parameterizations derived with pure or mixed aerosol samples, where the laboratory parameterization values can be overestimated by up to an order of magnitude. Several reasons have been proposed for the discrepancies between the parameterization and observation values, including the failure of parameterization to account for 1) the complex mixture of organic composition (Bertram et al., 2009; Mielke et al., 2013); 2) the "real" nitrate

suppression effect (Riedel et al., 2012; Morgan et al., 2015); and 3) the mixing states of the particles (Ryder et al., 2014; Wang X. et al., 2017). These results suggest the lack of comprehensive understanding of the $N_2O_5$ uptake and $ClNO_2$ production yield in various atmospheric environments around the world.

Most of the previous field studies of $N_2O_5$ uptake and $ClNO_2$ production have been conducted in the United States

(US) and Europe regions (Brown et al., 2009; Chang et al., 2016). Direct field investigation of the $N_2O_5$ heterogeneous processes in China is very limited. Pathak et al. (2009, 2011) analyzed the aerosol composition and suggested that the accumulation of fine $NO_3^-$ aerosol in downwind of Beijing and Shanghai was due to significant $N_2O_5$ heterogeneous



reactions. Wang et al. (2013) linked the observed $NO_3^-$ with the precursors of $N_2O_5$ (i.e., $NO_2$ and $O_3$) in urban Shanghai and suggested that the $N_2O_5$ heterogeneous uptake dominated $NO_3^-$ formation on polluted days. However, field measurements of $N_2O_5$ and $ClNO_2$ were not available until recently at several sites in southern and northern China (Tham et al., 2014; Wang Z. et al., 2017).

In the summer of 2014, a field campaign was carried out to investigate $ClNO_2$ and $N_2O_5$ at a semirural ground site at Wangdu in polluted northern China (Tham et al., 2016). Elevated levels of $ClNO_2$ up to 2070 pptv, but relatively low values of $N_2O_5$ (430 pptv), were observed on most nights at this site, and heterogeneous processes have been shown to have a significant effect on the following day's radical and ozone production at the site (Tham et al., 2016). Yet, the factors that

drive the $N_2O_5$ heterogeneous uptake and $ClNO_2$ production yield remain unclear. In this study, we further analyze the dataset to investigate this topic. We first derive values for $\gamma(N_2O_5)$ and $\phi$ with the measurement data and then compare the values obtained in the field with various parameterizations derived from the laboratory studies. With the aid of the aerosol composition and meteorological measurements, we illustrate the factors that drive or influence the variations in $\gamma(N_2O_5)$ and $\phi$ at Wangdu. The values for $\gamma(N_2O_5)$ and $\phi$ obtained here are also compared with field results from the literature to provide

an overview of the $N_2O_5$-$ClNO_2$ heterogeneous process observed in various environments around the world.

## 2 Methods

The measurement site (38.66°N, 115.204°E) is located at a semirural area in Wangdu county of Hebei province, in the

northern part of China. The Wangdu site is situated within the agricultural land but is bounded by villages and towns. Beijing (the national capital) is about 170 km to the northeast, Tianjin is about 180 km to the east, Shijiazhuang is about 90 km to the southwest, and Baoding is ~33 km to the northeast. Dozens of major coal-fired power stations are also located in the region. During the study period, frequent biomass burning activity was observed in the surrounding regions. Analysis of the air masses' back trajectories showed that the sampling site was frequently affected by these surrounding anthropogenic sources.

Details on the sampling site and the meteorological conditions during the campaign can be found in Tham et al. (2016).

In this study, $N_2O_5$ and $ClNO_2$ were measured with an iodide chemical mass ionization mass spectrometer (CIMS), with which the $N_2O_5$ and $ClNO_2$ were detected as the iodide-cluster ions of $I(ClNO_2)^-$ and $I(N_2O_5)^-$, similar to those outlined by Kercher et al. (2009). The detection principles, calibration, and inlet maintenance were described in detail in our previous

studies (Wang T. et al., 2016; Tham et al., 2016). The CIMS measurement at the Wangdu site was performed from 20 June to 9 July 2014. A corona discharge ion source setup (for generation of iodide primary ions) was used in the CIMS measurement from 20 to 26 June 2014 with a detection limit of 16 pptv for $N_2O_5$ and 14 pptv for $ClNO_2$ (3$\sigma$; 1 min-averaged data) but was replaced by a radioactive ion source from 27 June 2014 until the end of the study with a detection limit of 7



pptv for $N_2O_5$ and 6 pptv for $ClNO_2$ ($3\sigma$; 1 min-averaged data). The overall uncertainty of the CIMS measurement was estimated to be ±25%, with a precision of 3%.

The present study was supported by other auxiliary measurements of aerosol and trace gases. Trace gases including NO, $NO_2$, $O_3$, $SO_2$, CO, and total odd nitrogen ($NO_y$) were measured with online gas analyzers (Tan et al., 2017). A gas aerosol collector–ion chromatography system was used to measure the ionic compositions of $PM_{2.5}$, including $NO_3^-$, $Cl^-$, $SO_4^-$, $NH_4^+$, and gas-phase $HNO_3$ (Dong et al., 2012). The particle surface area concentrations ($S_a$) were calculated based on the wet ambient particle number size distribution predicted from the size-resolved kappa-Köhler function determined from real-time measurement with a high humidity tandem differential mobility analyzer (Hennig et al., 2005; Liu et al., 2014). Hydroxyl radical (OH) was measured with the laser-induced fluorescence technique (Tan et al., 2017). Volatile organic compounds including methane, $C_2$-$C_{10}$ hydrocarbons, formaldehyde, and oxygenated hydrocarbons and acetonitrile ($CH_3CN$) were measured with a cavity ring–down spectroscopy technique instrument, an online gas chromatograph equipped with a mass spectrometer and a flame ionization detector, a Hantzsch fluorimetric monitor, and a proton-transfer-reaction mass spectrometer, respectively (Yuan et al., 2010; Wang et al., 2014). Meteorological data including the wind profile, relative humidity (RH), and temperature were measured with an ultrasonic anemometer and a weather station on a 20m tower. Detailed descriptions of the instrumentation and observations of the aerosols, trace gases, and meteorological parameters at Wangdu can be found in other publications (e.g., Wang Y. et al., 2016; Min et al., 2016; and Tham et al., 2016).

## 3 Results and Discussion

### 3.1 Nocturnal heterogeneous $N_2O_5$ reaction at Wangdu

Figure 1 illustrates the time series of $NO_x$, $O_3$, $N_2O_5$, $ClNO_2$, particulate $NO_3^-$, $S_a$, the calculated production rate of $NO_3$, and the lifetime of $N_2O_5$ observed at Wangdu between 20 June and 9 July 2014. Abundance of $NO_x$ and $O_3$ was observed at night-time (20:00 to 05:00 local time); with average night-time mixing ratios of 21 and 30 ppbv, respectively. The elevated night-time $NO_x$ and $O_3$ levels led to the active production of $NO_3$, with an average production rate of $NO_3$ ($=k_{O3+NO2}[NO_2][O_3]$) of 1.7 ppb h$^{-1}$ and a maximum level of 8.3 ppb h$^{-1}$ for the entire campaign. Even with the rapid production of $NO_3$ and the high $NO_2$ level at night, the observed $N_2O_5$ concentrations were typically low (i.e., average nighttime concentration of 34 ± 14 pptv). The low $N_2O_5$ value is consistent with the short steady-state lifetime of $N_2O_5$ ($\tau(N_2O_5)$) for the study period, ranged from 0.1 to 10 min, suggesting that the direct loss of $N_2O_5$ via heterogeneous reaction and/or indirect loss of $N_2O_5$ via decomposition to $NO_3$ (i.e., reactions of $NO_3$ with NO and volatile organic compounds [VOCs]) were rapid in this region. The good correlation between the night-time levels of $ClNO_2$ and fine particulate $NO_3^-$ (the products of heterogeneous reactions of $N_2O_5$ via R3 and R4, respectively), with a coefficient of determination ($r^2$) of greater than 0.6 on most nights, provides field evidence of active $N_2O_5$ heterogeneous uptake processes in this region.



### 3.2 Estimation of $N_2O_5$ uptake coefficient and $ClNO_2$ production yield

The consistent trends and clear correlation between $ClNO_2$ and fine particulate $NO_3^-$ could be used to quantify $N_2O_5$ heterogeneous uptake following the method described by Phillips et al. (2016). The uptake coefficient of $N_2O_5$, $\gamma$ ($N_2O_5$),

was estimated based on the production rate of $ClNO_2$ ($pClNO_2$) and the nitrate formation rate ($pNO_3^-$) from the following equation (4).

$$\gamma(N_2O_5)=\frac{2(pClNO_2+pNO_3^-)}{C_{N2O5}S_a[N_2O_5]} \qquad \text{(Eq 4)}$$

The yield of $ClNO_2$ was determined from the regression analysis of $ClNO_2$ versus particulate $NO_3^-$ (Wagner et al., 2012; Riedel et al., 2013). The slope ($m$) from the regression plot was fitted into equation (5) to obtain the $\phi$.

$$\phi = \frac{2m}{1+m} \qquad \text{(Eq 5)}$$

The concentrations of $ClNO_2$, $N_2O_5$, particulate $NO_3^-$, and other related data used for this analysis were averaged or interpolated into 10 min. This analysis assumes 1) that the air mass is stable and losses of $ClNO_2$ and $NO_3^-$ are insignificant within the duration of analysis; 2) that the $NO_3^-$ produced via $N_2O_5$ heterogeneous uptake remains in the particle phase and does not significantly degas as $HNO_3$; and 3) that the night-time production of $NO_3^-$ through the net $HNO_3$ uptake to aerosols

is not important compared to that formed via $N_2O_5$ heterogeneous uptake.

      The limitation of this method is that it cannot predict the $\gamma(N_2O_5)$ with negative changes in the concentrations of $ClNO_2$ or particulate $NO_3^-$, which may be a result of differences in the origin or age of the air mass. In accordance with this limitation and with assumption (1) above, we carefully select plumes during the night-time that have meet the following

criteria for the analysis: shorter periods of data, usually between 1.5 and 4 hours, with concurrent increases in $ClNO_2$ and $NO_3^-$. The wind conditions and air mass age in the plume, represented by the ratios of $NO_x$ to $NO_y$, were relatively stable, and no drastic changes were seen in other variables such as the particle surface area, RH, or temperature. In addition, the concentration of NO in the plume must be relatively constant as the presence of a transient NO plume may affect the concentration of $N_2O_5$, which can bias the estimation of $\gamma(N_2O_5)$. Figure 2 shows two examples of relatively constant

conditions of relevant chemical composition and environmental variables, together with a plot of $ClNO_2$ versus particle $NO_3^-$ for the night. It should also be noted that partitioning of $NO_3^-$ to gas-phase $HNO_3$ and the contribution of particulate $NO_3^-$ from other sources, like the reaction of OH with $NO_2$ and the oxidation of VOCs by $NO_3$, can bias the values predicted for $\gamma(N_2O_5)$ and $\phi$.

To check the validity of assumptions (2) and (3) above, we also calculated the production rate of $NO_3^-/HNO_3$ via reaction of $OH+NO_2$ ($=k_{OH+NO2}[OH][NO_2]$) and $NO_3+VOC$ ($=\Sigma_i k_i[VOC_i][NO_3]$), as shown in the average diurnal profiles of related species in Figure 3. It is clear that particulate $NO_3^-$ was the dominant species during the night-time at Wangdu (Figure 3b). The strong correlation between $ClNO_2$ and particulate nitrate during the night indicates that the heterogeneous process

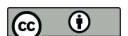



of $N_2O_5$ was the dominant source of particulate nitrate. Moreover, the production rate of $HNO_3$, as calculated from the gas-phase reactions of $OH+NO_2$ and $NO_3+VOC$, shows a decreasing trend towards the night (Figure 3c), and the combination of these rates on average is only about one-third of the average $pNO_3^-$ during the night. The increase in night-time $NO_3^-$ was also accompanied by an increase in ammonium ($NH_4^+$), which suggests that the repartition process to form ammonium nitrate was efficient, thus limiting the release of $HNO_3$ (Figure 3d). These results support the validity of the above assumptions and the determination of uptake and yield in this analysis.

With these methods and selection criteria, we can derive $\gamma(N_2O_5)$ and $\phi$ for 10 night-time plumes. Table 1 shows the estimated $N_2O_5$ uptake coefficients and $ClNO_2$ yields at Wangdu together with the errors that account for the scattering of data in the analysis and uncertainty from the measurement of $N_2O_5$, $ClNO_2$, aerosol surface area, and particulate $NO_3^-$. The estimated $\gamma(N_2O_5)$ values ranged from 0.006 to 0.034, with a median value of 0.023. A large variability was found in $\phi$ (range, 0.07 to 1.04). The relatively larger $\gamma(N_2O_5)$ and $\phi$ values observed on the night of 20–21 June are consistent with the observation of the highest $ClNO_2$ concentration, whereas the lower $\gamma(N_2O_5)$ and $\phi$ values on the night of 28–29 June explain the observation of the elevated $N_2O_5$ and small $ClNO_2$ mixing ratios (c.f. Figure 1). The observed $\gamma(N_2O_5)$ and $\phi$ values at Wangdu were compared with literature values derived from previous field observations in various locations in North America, Europe, and China, as summarized in Figure 4 and Table 2. The variable values of $\gamma(N_2O_5)$ in this study fall in the range of $\gamma(N_2O_5)$ (<0.001 to 0.11) and $\phi$ (0.01–1.38) reported around the world. The values are also within the range of $N_2O_5$ uptake coefficients and $ClNO_2$ yields determined in regions of China regions ($\gamma(N_2O_5)$ = 0.004–0.103; $\phi$ = 0.01–0.98), which are in the middle to upper end of the values reported around the world. The observed significant $ClNO_2$ concentrations and high yields of $\phi$ here, consistent with other studies at inland sites (c.f. Table 2), also point to the fact that $ClNO_2$ production can be efficient in regions far from the oceanic source of chloride and further highlight the important role of anthropogenic chloride emissions in the chlorine activation process and the next-day's photochemistry. The question arises here is what drive the large variability in the $\gamma(N_2O_5)$ and $\phi$ at Wangdu.

### 3.3 Factors that control the $N_2O_5$ uptake coefficient

Heterogeneous uptake of $N_2O_5$ is governed by various factors, including the amount of water and the physical and chemical characteristics of the aerosols (Chang et al., 2011; Brown and Stutz, 2012). To gain better insight into the factors that drive the $N_2O_5$ heterogeneous uptake, the determined $\gamma(N_2O_5)$ values were compared with those predicted from complex laboratory-derived parameterizations, and their relationships with the aerosol water content and aerosol compositions observed at Wangdu were examined.

The parameterization of $N_2O_5$ uptake coefficient derived from Bertram and Thornton (2009) ($\gamma_{B\&T}$) considers the amount of nitrate, chloride, and water in the aerosol as the controlling factors and can be calculated with equation (6):





$$\gamma_{B\&T} = Ak\left(1 - \frac{1}{\left(\frac{k_{R3}[H_2O](l)}{k_{R2b}[NO_3^-]}\right) + 1 + \left(\frac{k_{R4}[Cl^-]}{k_{R2b}[NO_3^-]}\right)}\right) \qquad \text{(Eq 6)}$$

where $A$ is an empirical pre-factor calculated from the volume of aerosol ($V$), $S_a$, $c_{N2O5}$, and Henry's law coefficient of $N_2O_5$ ($A= 4/c_{N2O5} \times V/S_a \times H_{aq}$); $k = 1.15 \times 10^6 - (1.15 \times 10^6)^{exp(-0.13[H2O])}$; $k_{R3}/k_{R2b} = 0.06$; and $k_{R4}/k_{R2b} = 29$. The concentration of aerosol liquid water ([$H_2O$]) used in this study was estimated from the E-AIM model IV with inputs of measured aerosol composition (http://www.aim.env.uea.ac.uk /aim/model4/model4a.php) (Wexler and Clegg, 2002), and the $V/S_a$ was taken from the field measurement at Wangdu. This $\gamma_{B\&T}$, however, does not account for the suppression of $\gamma(N_2O_5)$ from the organics, but it is frequently used with the parameterization formulated by Anttila et al. (2006), who treated the organic fraction in the aerosols as a coating, as given in equation (7) (e.g., Morgan et al., 2015; Phillips et al., 2016; Chang et al., 2016). The net uptake of $N_2O_5$ onto an aqueous core and organic coating ($\gamma_{B\&T+Org}$) can be determined by equation (8).

$$\gamma_{Org} = \frac{4\,RTH_{org}D_{org}R_c}{C_{N2O5}LR_p} \qquad \text{(Eq 7)}$$

$$\frac{1}{\gamma_{B\&T+Org}} = \frac{1}{\gamma_{B\&T}} + \frac{1}{\gamma_{Org}} \qquad \text{(Eq 8)}$$

Here, the $H_{org}$ is the Henry's Law constant of $N_2O_5$ for organic coating; $D_{org}$ is the solubility and diffusivity of $N_2O_5$ in the organic coating of thickness $L$; and $R_c$ and $R_p$ are the radii of the aqueous core and particle, respectively. The $L$ was calculated following the method in Reimer et al. (2009) with the assumption of hydrophobic organic coating (density, 1.27 g cm$^{-3}$) on the aqueous inorganic core (with a density of 1.77 g cm$^{-3}$). The $H_{org}D_{org}$ is equal to $0.03 \times H_{aq}D_{aq}$, where $H_{aq} = 5000$ M atm$^{-1}$ and $D_{aq} = 10^{-9}$ m$^2$ s$^{-1}$ (Chang et al., 2011 and references therein). In addition, Evan and Jacob (2005) proposed a simpler parameterization of $N_2O_5$ uptake on sulfate aerosol ($\gamma_{E\&J}$) as a function of temperature and RH, as given by equation (9).

$$\gamma_{E\&J} = (2.79\times10^{-4}+1.3\times10^{-4}\times RH - 3.43\times10^{-6}\times RH^2 + 7.52\times10^{-8}\times RH^3)\times10^{(4\times10^{-2}\times(T-294))} \qquad \text{(Eq 9)}$$

Figure 5 illustrates a comparison of field-derived $N_2O_5$ uptake coefficients with the values computed from the above parameterizations. The computed $\gamma_{B\&T}$ (red circle) ranged from 0.046 to 0.094 and was consistently higher than the field-derived $\gamma(N_2O_5)$ by up to a factor of 9. By accounting for the effects of organic coating on the $N_2O_5$ uptake coefficient via equations (7) and (8), the calculated $N_2O_5$ uptake coefficients (green circle in Figure 5) are significantly underestimated. Note that only six cases were available to compute the $\gamma_{B\&T+Org}$ due to the limited organic aerosols data for the study period. The $N_2O_5$ uptake coefficients computed from the parameterization suggested by Evan and Jacob (2005) are generally consistent with the field-derived $\gamma(N_2O_5)$ (as shown by blue circles). The different results from these parameterizations may suggest more complex aerosol composition or properties in the real ambient atmosphere than in the aerosol sample used in the laboratory study.

We then examine the relationships of the field-derived $\gamma(N_2O_5)$ with RH, water content, and aerosol compositions, as





illustrated in Figure 6. It can be seen in Figure 6a that the $\gamma(N_2O_5)$ has a clear correlation with the aerosol water content ($r^2 =$ 0.86; $p < 0.01$, $t$-test). The strong dependence of $\gamma(N_2O_5)$ on the aerosol water content was observed at RH lower than 80% or [$H_2O$] lower than 40 mol L$^{-1}$. The $\gamma(N_2O_5)$ then plateaus at about 0.032 when the RH exceeds 80%. This pattern is similar to the trends observed in laboratory studies for $N_2O_5$ uptake onto aqueous sulfate and malonic acid aerosols, in which the

$\gamma(N_2O_5)$ strongly increases with humidity at RH below 40-50% but becomes insensitive above this threshold (e.g., Hallquist et al., 2003; Thornton et al., 2003). The $\gamma(N_2O_5)$ at Wangdu shows a trend of decreasing with the concentration of $NO_3^-$ per volume of aerosol (see Figure 5b), which is similar to the results from the previous laboratory studies (Bertram and Thornton, 2009; Griffiths et al., 2009). However, we do not think that [$NO_3^-$] is the dominant limiting factor for $N_2O_5$ uptake at this site, as seen in the consistency of the $\gamma(N_2O_5)$ data points with the change in RH (in the color code of Figure 6b), the increasing

trend of $\gamma(N_2O_5)$ with the concentration of particulate nitrate in the air (c.f. Figure 6c), and the positive dependency of $\gamma(N_2O_5)$ on the molar ratio of [$H_2O$]/[$NO_3^-$] (c.f. Figure 6d), which reflect that the $N_2O_5$ uptake is more sensitive to the aerosol water content than to the $NO_3^-$, at least up to [$H_2O$]:[$NO_3^-$] of 20. The increase in ambient particulate nitrate is probably due to the faster $N_2O_5$ heterogeneous reaction. The $N_2O_5$ uptake does not show an increasing trend with the chloride-to-nitrate molar ratio, a pattern demonstrated in the laboratory result (Bertram and Thornton, 2009), but rather a

decrease for high [$Cl^-$]/[$NO_3^-$] ratios, and it also correlates with differences in RH (c.f. Figure 6e). There is a lack of correlation of $\gamma(N_2O_5)$ with the [Org]:[$SO_4^-$] observed in the ratio range of 0.5–1.2, indicating that the suppression of organics on the $N_2O_5$ uptake may be insignificant at Wangdu. These results are in line with the parameterization comparison results shown in Figure 5, reveal that the variation in the $N_2O_5$ uptake at Wangdu is not driven by the chemical properties of aerosols like $NO_3^-$, $Cl^-$, and organics, but rather that the RH or the aerosol water content plays a defining role in the $N_2O_5$

heterogeneous uptake.

The response of $N_2O_5$ uptake on the changes in RH is consistent with the changes in the sulfate ($SO_4^{2-}$) concentrations (see Figure 6g), which mainly determine the hygroscopicity of the aerosols and were found to be responsible for the particle growth at Wangdu (Wu et al., 2017). The hygroscopic growth of aerosols inferred by the RH (water uptake) can affect the

amount of water in the aerosol and the volume-to-surface area ratio ([$H_2O$]$V$/$S_a$). The good positive correlation of $\gamma(N_2O_5)$ with [$H_2O$]$V$/$S_a$ (see Figure 6h) suggests that the increased volume of aerosol, in particular the layer of aerosol water content, could allow efficient diffusion of $N_2O_5$ and solvation of $N_2O_5$ into $H_2ONO_2^+$ and $NO_3^-$ for further aqueous reactions, whereas a smaller volume of aerosol (less water content) may be easily saturated by $N_2O_5$ and then diffuse the $N_2O_5$ out of the aerosol, limiting the solvation of the $N_2O_5$ process and restricting $N_2O_5$ uptake. Several laboratory studies have demonstrated that an

increase in RH enhanced the particle aqueous volume and increased the bulk reactive $N_2O_5$ uptake on aqueous acids (i.e., malonic, succinic, and glutaric acid) and aqueous sulfate containing aerosols (Thornton et al., 2003; Hallquist et al., 2003). The increase in RH can also lower the viscosity of the aqueous layer in organic-containing aerosols, leading to greater diffusivity of $N_2O_5$ within the aerosol water layer, which ultimately increases the $N_2O_5$ uptake (Gržinic et al., 2015).



The strong dependency of N$_2$O$_5$ uptake upon the RH has not been clearly demonstrated in other field measurements. Field observations of γ(N$_2$O$_5$) in North America and Europe show any significant direct dependence of γ(N$_2$O$_5$) on RH but were strongly influenced by the aerosol composition (refer to the descriptions in Table 2). For instance, the flight measurements in Texas and London showed the independence of γ(N$_2$O$_5$) at RH from 34% to 90%, but the γ(N$_2$O$_5$) were

generally controlled by the amount of NO$_3^-$ and/or organic compounds (Brown et al., 2009; Morgan et al., 2015). A comparison of ground measurements in Seattle and Boulder showed variations in γ(N$_2$O$_5$) at various H$_2$O($l$) levels in the two places, but the RH alone was insufficient to describe their observed γ(N$_2$O$_5$) variability, and the organic composition of the aerosols was determined to have a dominant influences on γ(N$_2$O$_5$) (Bertram et al., 2009). Another study from a mountainous site in Germany reported no significant correlation of γ(N$_2$O$_5$) with aerosol compositions and only a weak dependence on

humidity (Phillips et al., 2016). However, field measurements at Jinan and Mt. Tai in northern China during the same season also showed a positive relationship of γ(N$_2$O$_5$) with an RH between 43% and 72% and aerosol water content of 31–65 mol L$^{-1}$, respectively (Wang X. et al., 2017; Wang Z. et al., 2017). The results of this study and previous reported results in the region may suggest that RH and aerosol water content are the important limiting factors for the N$_2$O$_5$ heterogeneous process in the polluted northern China. In summary, the more complex parameterizations considering nitrate, chloride, and the

organic coating cannot fully represent the variation of γ(N$_2$O$_5$) at Wangdu; instead, a simple parameterization that accounts only for temperature and RH appears to explain the variation in γ(N$_2$O$_5$) at Wangdu. It would be of great interest to determine whether such a phenomenon can be found in other places.

### 3.4 Factors that affect the ClNO$_2$ production yield

In addition to the uptake coefficients, the factors that influence the branching yield of ClNO$_2$ from the N$_2$O$_5$ heterogeneous uptake were also assessed. Figure 7a shows the scatter plot of the φ calculated from equation (3) versus the ClNO$_2$ yield derived from the Wangdu field data from equation (5) in Section 3.2. Generally, the φ$_{param.}$ shows less variability but was obviously overestimated in relative to the field-determined φ. Such discrepancy has also been observed

elsewhere (Thonton et al., 2010; Mielke et al., 2013; Riedel et al., 2013), including our recent observations in an urban site (Jinan) and a mountaintop site (Mt. Tai) in northern China, where the parameterized φ would be overestimated by up to two orders of magnitude (Wang X. et al., 2017; Wang Z. et al., 2017). Further analysis by linking the field-derived ClNO$_2$ yields with the aerosol water content (Figure 7b) and the Cl$^-$ content (Figure 7c) show a weak positive correlation ($r^2 = 0.34$) and a weak negative trend ($r^2 = 0.25$), respectively (from quadratic fitting). The weak correlations reflect that the ClNO$_2$ yield is

not solely controlled by the amount of water and chloride in the aerosol, as defined in the parameterization (see Eq. 3), and/or the existence of other nucleophiles that can compete with Cl$^-$ in reactions R3 and R4.

An interesting observation from Wangdu is that the field-derived φ shows a good decreasing trend ($r^2 = 0.65$ from quadratic data fitting) with the ratio of acetonitrile to carbon monoxide (CH$_3$CN/CO), which is an indicator of biomass





burning emission (Christian et al., 2003; Akagi et al., 2011), as illustrated in Figure 7d. The φ decreased at larger $CH_3CN/CO$ ratios, which corresponds to higher $[Cl^-]$ concentrations per volume of aerosols (c.f. Figure 7c) because biomass burning emits a significant level of chloride particles. The observations here may suggest that the φ is likely "suppressed" in air masses influenced by biomass burning, which were frequently observed during the study period (Tham et al., 2016), and is

consistent with the recent field observation of much lower concentrations of $ClNO_2$ during the bonfire event in Manchester compared to that after the event (Reyes-Villegas et al., 2017) and with a laboratory experiment which demonstrated that only a small amount (~10%) of reacted $N_2O_5$ was converted to $ClNO_2$ on the biomass-burning aerosols (Ahern et al., 2017). Another laboratory study showed that the $ClNO_2$ yield can be reduced by as much as 80% in the presence of aromatic organic compounds like phenol and humic acid in the aerosol (Ryder et al., 2015), and previous studies in China reported

abundant humic-like substances (e.g., aromatic organic compounds) in aerosols with a large contribution from biomass burning (Fu et al., 2008). Therefore, the frequently observed influence of biomass burning at the Wangdu site during the campaign could in part explain the lower φ values and the discrepancy between observation and parameterization. Other factors, such as the nonuniform distribution of chloride within the aerosol, might also contribute to the overestimation of φ from the parameterization (Riedel et al., 2013). More studies are needed to investigate the effects of biomass burning

emissions on the heterogeneous process.

## 4 Summary and conclusions

We present an in-depth analysis of the $N_2O_5$ uptake coefficient and $ClNO_2$ yield in a polluted northern China environment

during the summer of 2014. Large variations in the levels of $\gamma(N_2O_5)$ and φ were observed during the study, ranging from 0.006 to 0.034 and from 0.07 to 1.04, respectively. A comparison between the $\gamma(N_2O_5)$ values derived from the field and the parameterizations that considered the nitrate and chloride levels and the hydrophobic organic coating showed poor agreement, suggesting more complex influences of ambient aerosol properties at the site than with the pure or mixed samples used in the laboratory. The $\gamma(N_2O_5)$ values at Wangdu were found to have a clear dependence on the RH and the aerosol

water content, a phenomenon that has not been reported in previous field studies in the United States or Europe. The parameterization that explicitly considers the dependence on RH showed better agreement with the field-derived $\gamma(N_2O_5)$ compared to the more complex formulation that considers the aerosol composition. The $ClNO_2$ yield from the parameterization is generally overestimated when compared to the field derived values. The observed φ was found to be "suppressed" in the air masses influenced by biomass burning even though abundant aerosol chloride was present. The

results of this study point to the need for more field and laboratory studies to obtain realistic parameterization of the heterogeneous processes of $N_2O_5$ and $ClNO_2$ to better simulate the ozone and aerosol production in air quality models in regions of China with high $NO_x$ emissions.



**Acknowledgment.** The authors thank Steven Poon, Qiaozhi Zha, Zheng Xu and Hao Wang for the logistics support, to Liming Cao and Lingyan He for providing the aerosol mass spectrometer data and to Li Zhang for scientific discussion. This work was funded by the National Natural Science Foundation of China (91544213, 41505103 and 41275123), National Key Research and Development Program of China (2016YFC0200500), PolyU Project of Strategic Importance (1-ZE13) and

Research Institute for Sustainable Urban Development (RISUD). The Peking University team acknowledges support from the National Natural Science Foundation of China (21190052) and the Strategic Priority Research Program of the Chinese Academy of Sciences (XDB05010500). The Leibniz Institute for Tropospheric Research team acknowledges funding from the Sino German Science Center (No. GZ663).

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





**Table 1: N$_2$O$_5$ uptake coefficients and ClNO$_2$ production yields from 10 selected plumes at Wangdu during the summer of 2014. The uncertainty (±) of the γ(N$_2$O$_5$) and ϕ was estimated by the geometric mean of uncertainty from the scattering of the data plots and uncertainty from measurement of N$_2$O$_5$, ClNO$_2$, aerosol surface area, and particulate NO$_3^-$.**

| Plume | Date (Time) | γ(N$_2$O$_5$) | ϕ(ClNO$_2$) |
|---|---|---|---|
| 1 | 20 June (23:00) – 21 June (01:20) | 0.029 ± 0.011 | 0.54 ± 0.19 |
| 2 | 21 June (03:30 – 05:00) | 0.032 ± 0.015 | 1.04 ± 0.35 |
| 3 | 24 June (20:30 – 22:10) | 0.014 ± 0.008 | 0.10 ± 0.05 |
| 4 | 24 June (22:30) – 25 June (00:00) | 0.027 ± 0.009 | 0.38 ± 0.12 |
| 5 | 27 June (20:40) – 28 June (00:00) | 0.011 ± 0.007 | 0.15 ± 0.09 |
| 6 | 28 June (22:30) – 29 June (00:40) | 0.006 ± 0.002 | 0.20 ± 0.06 |
| 7 | 29 June (22:00) – 30 June (01:20) | 0.015 ± 0.005 | 0.28 ± 0.12 |
| 8 | 30 June (21:10) – 1 July (00:10) | 0.019 ± 0.007 | 0.18 ± 0.06 |
| 9 | 5 July (00:30) – 5 July (02:30) | 0.032 ± 0.019 | 0.45 ± 0.27 |
| 10 | 5 July (23:40) – 6 July (02:00) | 0.034 ± 0.014 | 0.07 ± 0.03 |



**Table 2**: **Summary of field-observed N$_2$O$_5$ uptake coefficient and ClNO$_2$ yield from previous studies.**

| Location | Environment | γ(N$_2$O$_5$) | φ | Descriptions | Reference |
|---|---|---|---|---|---|
| **North America** | | | | | |
| New England, US | Coastal + Inland | 0.001–0.017 | *n.a* | Aircraft measurement (below 1500 m). γ(N$_2$O$_5$) is higher in elevated sulfate region. | Brown et al., 2006 |
| Coast of Texas, US | Coastal | *n.a* | 0.10–0.65 | Shipborne measurement. Influenced by urban outflow. | Osthoff et al., 2008 |
| Texas, US | Coastal + Inland | 0.0005–0.006 | *n.a* | Aircraft measurement (below 1000 m). γ(N$_2$O$_5$) was independent of humidity (RH, 34% to 85%) and aerosol compositions. | Brown et al., 2009 |
| Seattle, US | Coastal | 0.005–0.04 | *n.a* | Urban/suburban environment. γ(N$_2$O$_5$) was enhanced with higher RH but has strong correlation with organic-to-sulfate ratio. | Bertram et al., 2009 |
| Calgary, Canada | Inland | 0.02 | 0.15 | Ground urban area. Influenced by anthropogenic activities within the urban area. | Mielke et al., 2011 |
| La Jolla, US | Coastal | 0.001–0.029 | *n.a* | Polluted coastal site. γ(N$_2$O$_5$) was suppressed by nitrate. | Riedel et al., 2012 |
| Coast of Los Angeles, US | Coastal | *n.a* | 0.15–0.62 | Shipborne measurement. Influenced by land-sea breeze. | Wagner et al., 2012 |
| Pasadena, US | Coastal | γφ = 0.008 (average) | | Ground measurement during the California Nexus 2010 campaign. γφ was enhanced by submicron chloride, but suppressed by organic matter and liquid water content. | Mielke et al., 2013 |
| Boulder, US | Inland | 0.002–0.1 | 0.01–0.98 | Tower measurement (0-300 m) downwind of urban city. γ(N$_2$O$_5$) dependence on nitrate. Higher φ in coal combustion plume. | Wagner et al., 2013 Riedel et al., 2013 |
| **Europe** | | | | | |
| London | Coastal + Inland | 0.01–0.03 | *n.a* | Aircraft measurement (500-1000 m). γ(N$_2$O$_5$) was independent of humidity (RH, 50% to 90%) but dependent on nitrate loading. | Morgan et al., 2015 |
| Kleiner Feldberg | Inland | 0.004–0.11 | 0.029–1.38 | Semirural mountain-top site in SW Germany (825 m above sea level). γ(N$_2$O$_5$) was independent of aerosol compositions but has a weak dependence on humidity. | Phillips et al., 2016 |
| **China** | | | | | |



| | | | | | |
|---|---|---|---|---|---|
| Hong Kong | Coastal | 0.004–0.021 | 0.02–0.98 | Rural mountain-top site in southern China (957 m above sea level). Influenced by pollution from urban area. | Brown et al., 2016 <br> Yun et al., 2018 |
| Jinan | Inland | 0.042–0.092 | 0.01–0.08 | Urban-surface in polluted urban area of northern China. $\gamma(N_2O_5)$ showed positive dependence on RH. | Wang X. et al., 2017 |
| Mt. Tai | Inland | 0.021–0.103 | 0.17–0.90 | Mountaintop site in northern China (1465 m above sea level). Elevated $\gamma(N_2O_5)$ for high humidity (>80%) condition. Higher $\phi$ in coal-fired power plant plumes. | Wang Z. et al., 2017 |
| Beijing-urban | Inland | 0.025−0.072 | *n.a* | Polluted urban surface-site in northern China during early autumn. High $\gamma(N_2O_5)$ was related to high aerosol liquid water content. | Wang H. et al., 2017 |
| Beijing-rural | Inland | 0.012–0.055 | 0.50–1.00 | Rural surface site in northern of Beijing. Influenced by the outflow of the urban Beijing. | Wang H. et al., 2018 |
| Wangdu | Inland | 0.006–0.034 | 0.07–1.04 | Semirural and surface site in northern China. $\gamma(N_2O_5)$ has strong dependence on humidity and aerosol water content. Variable $\phi$ and lower values for cases influenced by biomass burning activities. | This study |

*n.a = no information available*



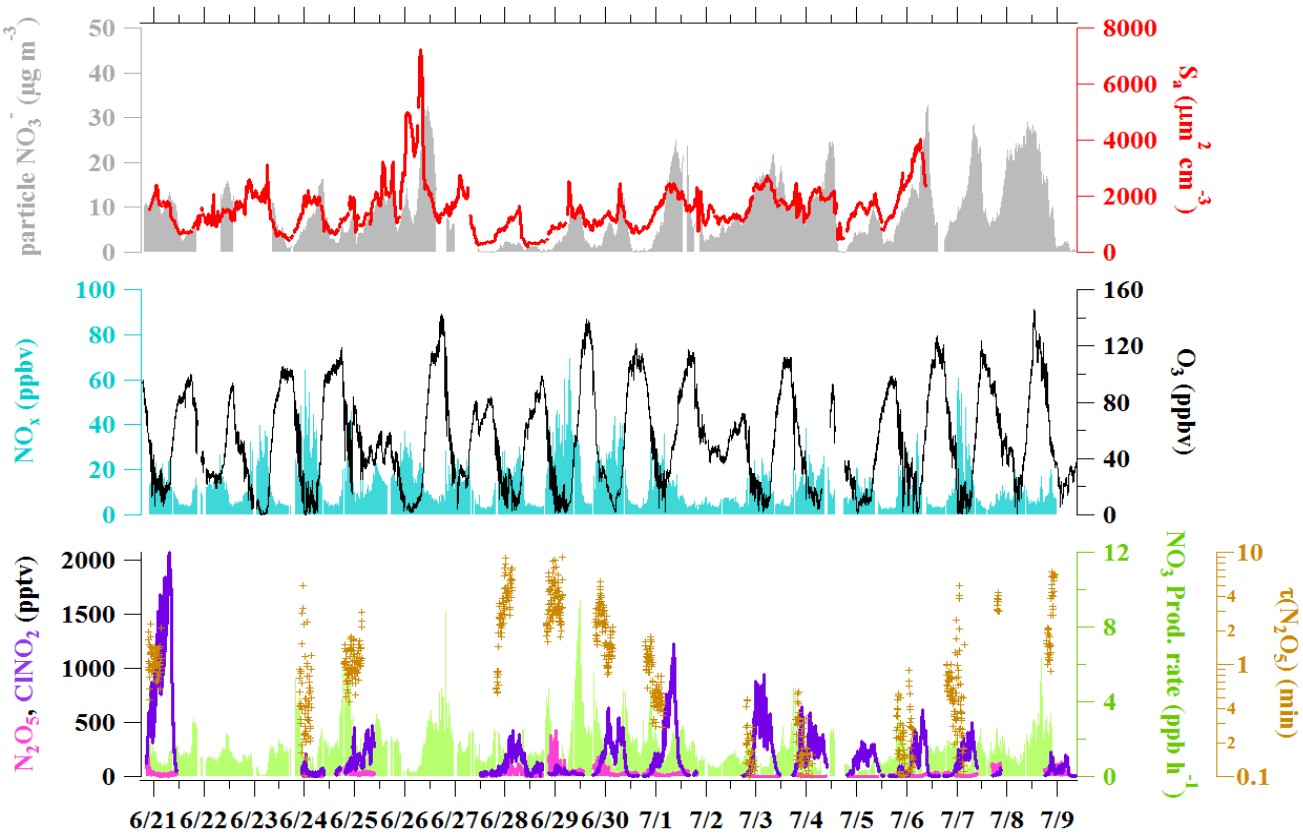

**Figure 1: Time series of N$_2$O$_5$, ClNO$_2$, NO$_3$ production rate, the steady-state lifetime of N$_2$O$_5$, O$_3$, NO$_x$, fine particulate NO$_3^-$, and S$_a$ data at Wangdu from 21 June to 9 July 2014. N$_2$O$_5$ and ClNO$_2$ are 1-min data, whereas the NO$_x$, O$_3$, NO$_3$ production rate and $\tau$(N$_2$O$_5$) are given as 5-min averages. The data for S$_a$ and fine particulate NO$_3^-$ are in 10-min and 30-min time resolutions, respectively. The data gaps were caused by technical problems, calibrations, or instrument maintenance.**





**Figure 2: Example of the accumulation of ClNO$_2$ and particulate NO$_3^-$ concentrations during the relatively constant condition of relevant chemical compositions and environmental variables observed for (a) Plume 1 on 20-21 June 2014 and (c) Plume 7 on 29-30 June 2014. Scatter plots of ClNO$_2$ versus particulate NO$_3^-$ to estimate the ClNO$_2$ yield (φ) for these two cases are shown in (c) and (d).**





**Figure 3: Diurnal variations of (a) N₂O₅ and ClNO₂; b) particulate NO₃⁻ and gas-phase HNO₃ (c) gas-phase production rate of NO₃⁻/HNO₃ via reaction of OH+NO₂ and NO₃+VOC; and (d) concentrations of NH₄⁺ in relation to particulate NO₃⁻.**







**Figure 4: Comparison of field-observed N$_2$O$_5$ uptake coefficient and ClNO$_2$ yield from previous studies. Sticks represent the range of the reported values, and cubes represent the median or average values reported in these measurements. The corresponding references are listed in Table 2.**



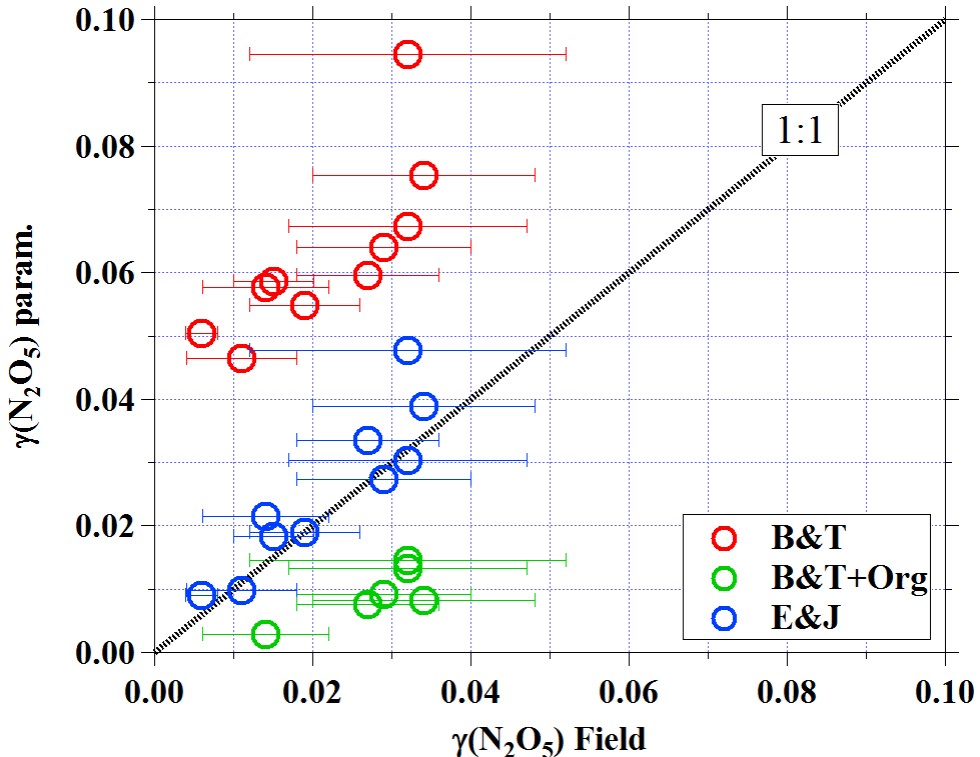

**Figure 5: Comparison of field-derived N$_2$O$_5$ uptake coefficients with values computed from different parameterizations. The dashed line represents 1:1, and the error bars show the uncertainty of γ(N$_2$O$_5$) derived from the field.**





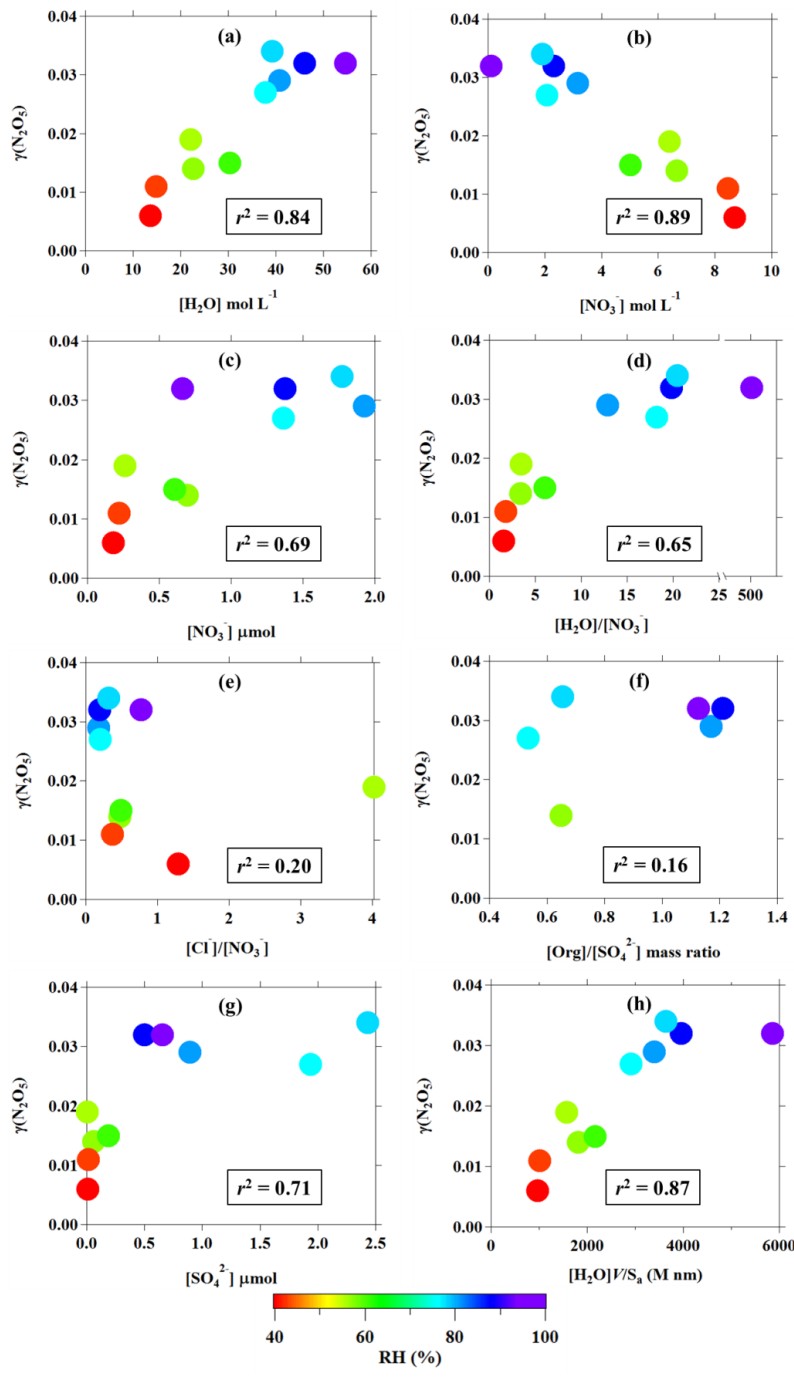

**Figure 6**: **Relationship between field-derived γ($N_2O_5$) and a) aerosol water content (mol per volume of aerosol); b) nitrate concentration per volume of aerosol; c) particulate nitrate concentration (μmol m$^{-3}$ of air); d) $H_2O$ to $NO_3^-$ molar ratio; e) Cl$^-$ to $NO_3^-$ molar ratio; f) organic-to-sulfate mass ratio (data from aerosol mass spectrometer); g) concentration of $SO_4^{2-}$ (μmol m$^{-3}$ of air); and h) amount of water in aerosol multiplied by the volume-to-surface area ratio. Color code represents the ambient RH, and the value in the box is the best correlation coefficient obtained from curve fittings.**





**Figure 7**: Scatter plots for (a) yield derived from the field versus yield calculated from the parameterization (error bars represent the uncertainty of field-derived $\phi$, and black dotted line represents the 1:1 ratio); (b) field-derived yield versus aerosol water content; (c) field-derived yield versus chloride; and (d) field-derived yield versus $CH_3CN/CO$.