# Peer review of "Heterogeneous $N_2O_5$ uptake coefficient and production yield of $CINO_2$ in polluted northern China: Roles of aerosol water content and chemical composition"

_Atmospheric Chemistry and Physics, 2018_

## Referee Comment (RC1) · Anonymous Referee #2 · 26 May 2018

General Comments:

This paper reports an analysis of N2O5 uptake coefficients and ClNO2 yields from a polluted site in the North China Plain during a summer 2014 field intensive. The analysis finds variation of N2O5 uptake coefficients that is characteristic of data sets in other parts of the world. Comparisons between field determinations and laboratory based parameterizations, and between the determined uptake coefficients and other variables, shows that aerosol liquid water / relative humidity is a determining factor. This finding is in contrast to field studies in the U.S. and Europe.

ClNO2 yields are shown to be lower than current parameterizations based on the competition between chloride and liquid water, consistent with findings from other regions. The authors suggest ClNO2 suppression on biomass burning derived particles despite higher chloride content in these aerosol.

Overall, the paper adds to the growing database of these analysis and will be a valuable contribution to the literature. Publication is recommended after the authors address the following comments.

Specific comments:

Page 2, line 4: "yielding N2O5" rather than "yielding a N2O5".

Page 3, line 20: Also add Morgan et al., 2015 and McDuffie et al., 2018, to this list.

Page 3, line 22: sentence not clear. Does "laboratory parameterizations can be overestimated" mean that the observations are higher than or lower then the parameterizations?

Page 3, line 32: "NO3- aerosol downwind of" rather than "NO3- aerosol in downwind of"

Page 4, line 8: Is the quoted N2O5 a maximum or an average? Please specify.

Page 5, lines 25-26: Are the quoted average production rates of NO3 for nighttime only for for nighttime and daytime?

Page 5, equation 4: The method of Phillips et al. (2016) is referenced, but the method for calculating the production rates in the numerator in the right hand side of the equation is not specified for the data here. How are these quantities (pClNO2 and pNO3-) determined?

Page 5, line 33: Define "most nights" – how many nights had r2 > 0.6 for the stated correlation?

Page 6, line 14-15: Assumption 3 is not reasonable. HNO3 is in equilibrium with aerosols regardless of how it is produced. That is, HNO3 equilibration and N2O5 uptake are not separate processes, but tightly coupled ones. The assumption is more likely intended to state that N2O5 heterogeneous uptake during the night of observations is a larger source of total soluble nitrate (HNO3 plus NO3-) than soluble nitrate production from the preceding day, or that the correlation with ClNO2 is determined by the nighttime produced nitrate rather than the background that was present at sunset.

Page 6, line 19: remove the word "have"

Page 6, line 22: Is there a quantitative definition of "drastic changes" here? In other words, is the data filtering arbitrary, or done in a well-defined manner using characteristics of time rates of change.

Page 6, lines 26-28: See comment above. The partitioning of total nitrate between gas and particle phase is an important limitation, and it would be useful to define any quantitative information, such as an aerosol thermodynamic model, that would indicate where this partitioning is. The photochemical soluble nitrate production should be in the background of the correlation (i.e., the intercept) and so might not affect the results.

Text on pages 6-7 and Figure 3: Explain why there is significant OH + NO2 during the night. Explain how particulate NO3- production from NO3 + VOC is calculated. Many NO3 + VOC reactions produce organic nitrates rather than HNO3, so it is not clear how this source of HNO3 has been calculated based on the information given. The total production rate of NO3- is also referenced in the text but not shown in the figure. The differentiation between day and night in Figure 3 is not clear. Presumably the time axis is local time, not UTC? Please specify for clarity. The times of day and night should be shown, preferably with a shaded region to indicate night. Data for gas phase HNO3 are presented here for the first time. Why in the preceding analysis was ClNO2 only correlated against particulate phase NO3- if gas phase HNO3 is also available? The analysis should be done from the correlation between ClNO2 and total nitrate (HNO3

+ NO3-) since the two are in rapid equilibrium on the time scale of ClNO2 production through N2O5 uptake.

Page 7, line 22: "The question that arises" rather than "The question arises"

Page 7, line 33: "coefficients" rather than "coefficient"

Page 8, line 21: Figure 5 would be clearer if the field data were on the y-axis and the parameterization on the x-axis.

Page 9, line 22: "to changes in RH" rather than "on the changes in RH"

Page 9, lines 25-30: Is [H2O]V/Sa really independent of aerosol water itself? It seems that the effects discussed here and on the rest of page 9 can be determined from laboratory experiments under controlled conditions but not easily determined from field data. The authors should be careful to phrase this argument as consistent with laboratory data rather than a determination of these effects from field measurements.

Page 9, first paragraph: The major conclusion is that RH, and by extension the calculation of aerosol liquid water, was the determining factor for N2O5 uptake. In this context, it will be helpful to say more about the measurement of the wet aerosol surface area and its associated uncertainties, since wet aerosol surface area is often a difficult quantity to measure, and the measurement or calculation can itself introduce an RH dependence to the aerosol surface area measurement. The description in the methods section (Page 5, lines 7-9) is brief. A more comprehensive description of this measurement and statement of its potential dependence on RH, along with the uncertainty in the aerosol surface area, is needed.

Page 10, line 22, Figure 7a: As for figure 5, this would be clearer with the field data on the y-axis. All other plots in figure 7 have field data on the y-axis, and the same should be done for figure 7.

Page 10, line 24: remove the word "in". Also "Such a discrepancy" rather than "Such discrepancy".

Page 10, line 29: What is meant by "from quadratic fitting"? Is there a polynomial fit that should appear in Figure 7?

Page 10, line 33: Remove the word "good" or else replace by something more specific, such as "statistically significant", if appropriate. Also, the term "quadratic data fitting" appears again here without explanation or a displayed fit.

---

## Referee Comment (RC2) · Anonymous Referee #1 · 24 Jun 2018

Tham et al report N2O5 uptake coefficients and ClNO2 yields based on measurements of N2O5, ClNO2, and PM2.5 aerosol size distribution and composition at Wangdu in the summer 2014. The N2O5 uptake coefficients and ClNO2 yields were estimated based on observed production of ClNO2, (bulk) particulate nitrate, and in situ N2O5 concentration and aerosol surface area. These observed values are compared with predictions from several literature parameterizations. The authors show that $\gamma$(N2O5) increases with relative humidity (and aerosol liquid water content) and decreases with increasing particulate nitrate content. ClNO2 yields were variable and appeared to

show a decreasing trend in the presence of BB aerosol.

The paper is written well and will be a useful addition to the literature once the authors have satisfactorily addressed the comments below.

Major comments 1) Bulk aerosol properties are used in the analysis to calculate, for example, aerosol liquid water content at equilibrium, N2O5 uptake, and ClNO2 yield. In reality, however, the aerosol will consist of particles that have varying degrees of external (and internal) mixing. This may be particularly important for N2O5 to ClNO2 conversion, which takes place very efficiently on (supermicron) sea salt derived aerosol or in certain power plant plumes, but hardly at all on secondary aerosol that contains little chloride. Furthermore, the conversion of N2O5 to ClNO2 occurs mainly on the aerosol surface and not in the bulk. The authors should add more discussion on the limitations arising from the use of bulk aerosol properties in their analysis.

2) A major limitation, which unfortunately has become quite common in the literature, is to perform analysis with in situ variables (i.e., ClNO2 and N2O5 concentrations) and with variables that will integrate over the air mass's history, such as aerosol nitrate, and then to assume that upwind conditions were similar. This is a major assumption, of course, and many preceding papers spent a lot of time justifying it. It may be useful to add more discussion on what the upwind air masses typically would experience prior to observation (e.g., absence/presence of local sources etc.) at Wangdu.

3) In part because of (2), data were selected in the analysis. While the selection criteria are stated, it is in principle worrisome and may lead to selection bias. Can anything be said about the data that were excluded from analysis? For example, what fraction of the data were excluded, and can you give an indication as to what happens in terms of N2O5 to ClNO2 conversion during those periods - were the mixing ratios of ClNO2 high or low, and was the uptake of N2O5 fast or slow? Could these data be analyzed and added with a lighter shade to some of the Figures?

4) The conversion of N2O5 to ClNO2 is often stratified vertically, with usually rapid

N2O5 losses at the surface, and higher ClNO2 production rates aloft. How does stratification / vertical mixing affect the analysis?

Minor comments

page 1 / line 19 - replace "10" with "ten"; state on what basis cases were selected and how the N2O5 uptake coefficients and ClNO2 yields were estimated

line 21 - grammar: "an average", but then two values (one for N2O5 and one for ClNO2) are given; formatting for the ranges given in brackets is not consistent; The authors should state their estimated errors of the "observed" N2O5 and ClNO2 uptake parameters here.

line 25 - "by the amount of water in the aerosol, a phenomenon that differs from other field observations". Most models and the Bertram/Thornton parameterization (Eq 3) that contains a water term and would have been included in other field studies. Is the author's statement then really true?

line 26 - "Laboratory-derived parameterization also overestimated the ClNO2 yield." Please correct the grammar here.

pg 3/ line 11 - "450" Roberts et al. Geophys. Res. Lett., 36, L20808, 10.1029/2009GL040448, 2009 give a much larger value here; consider adding a second set of ÏŢparm calculations with the Roberts et al. value and add to Figure 7a.

page 4 / line 11 "We first derive values for $\gamma$(N2O5) and ÏŢ with the measurement data". Please state briefly here how this is done.

page 5. Please add a table summarizing the various measurements made. Without one, statements such as "Volatile organic compounds including methane, C2-C10 hydrocarbons, formaldehyde, and oxygenated hydrocarbons and acetonitrile (CH3CN) were measured with a cavity ring–down spectroscopy technique instrument, an online gas chromatograph equipped with a mass spectrometer and a flame ionization detector, a Hantzsch fluorimetric monitor, and a proton-transfer-reaction mass spectrometer, respectively" are unnecessarily confusing.

line 28 - "steady-state" Brown et al. (J. Geophys. Res., 108, 4539, 10.1029/2003JD003407, 2003) showed that the time to achieve a steady state can be substantial, especially in polluted conditions. Have the authors verified (e.g., through box model simulations) that the steady-state approximation is valid?

page 7 / line 23 "what drive" Grammar (either "what drives" or "what factors drive")

page 8 / the "observed" $\gamma$(N2O5) is really an aggregate vale for N2O5 uptake on the entire aerosol distribution

line 2 - [H2O], [NO3-], and [Cl-] will likely be functions of aerosol size; please add a disclaimer that this calculation assumes that they are not, and that the predicted gamma values may be biased as a result.

Out of curiosity - is it possible that ClNO2 is produced mainly on sea salt aerosol at Wangdu?

line 4 - the E-AIM allows for inclusion of organics, which would alter the liquid water content (maybe). Has this been considered

line 12 what values of Rc and Rp were used in the B&T+org calculation, and are these values realistic for this comparison? (see also major comment 2).

pg 9 / line 16 - sulfate should be doubly charged

pg 10 / factors that affect ClNO2 yield - this is an interesting paragraph, but I am a bit skeptical about what appear to be low field yields.

Have the authors considered that the lack of agreement may be due to breakdown of the assumptions going into the calculation (uneven distribution of chloride throughout the aerosol, for example)?

page 22 - Please increase the font size on figures 2a and 2c (they are too small).

In Figures 2b and 2d, do the axis intercepts allow an assessment of how much aerosol nitrate is derived from daytime vs nighttime chemistry?

---

## Author Comment (AC2) · 14 Aug 2018

Tham et al report N2O5 uptake coefficients and ClNO2 yields based on measurements of N2O5, ClNO2, and PM2.5 aerosol size distribution and composition at Wangdu in the summer 2014. The N2O5 uptake coefficients and ClNO2 yields were estimated based on observed production of ClNO2, (bulk) particulate nitrate, and in situ N2O5 concentration and aerosol surface area. These observed values are compared with predictions from several literature parameterizations. The authors show that γ(N2O5) increases with relative humidity (and aerosol liquid water content) and decreases with increasing particulate nitrate content. ClNO2 yields were variable and appeared to show a decreasing trend in the presence of BB aerosol.

The paper is written well and will be a useful addition to the literature once the authors have satisfactorily addressed the comments below.

**Response**: We thank the reviewer for his/her attention to this manuscript. We have made all of the suggested changes and clarifications. The reviewer's comments are in black and our responses are in blue, and the changes in the manuscript are in *italic*.

**Major comments:**
1) Bulk aerosol properties are used in the analysis to calculate, for example, aerosol liquid water content at equilibrium, N2O5 uptake, and ClNO2 yield.
In reality, however, the aerosol will consist of particles that have varying degrees of external (and internal) mixing. This may be particularly important for N2O5 to ClNO2 conversion, which takes place very efficiently on (supermicron) sea salt derived aerosol or in certain power plant plumes, but hardly at all on secondary aerosol that contains little chloride. Furthermore, the conversion of N2O5 to ClNO2 occurs mainly on the aerosol surface and not in the bulk. The authors should add more discussion on the limitations arising from the use of bulk aerosol properties in their analysis.

**Response:** We agree with the reviewer, and we are aware of the possible bias resulted from the assumption on bulk reaction and the lack of aerosol mixing information. Current parameterizations (e.g., Bertram and Thornton, 2009) were based on the laboratory experiments with pure or internally mixed aerosols, and derived the uptake dependence on the bulk composition of wet aerosols. The E-AIM model used to calculate the aerosol liquid water content at equilibrium is also based on bulk aerosol composition. Thus, most of the recent studies and parameterizations did not specifically consider the mixing states of the aerosols, which may largely affect the $N_2O_5$ uptake and $ClNO_2$ yield on complex ambient aerosols. To clarify, we have added more information on the method and more discussion of the limitations in different parts of the revised text, as follows,

In introduction:

[revised manuscript text omitted]

2) A major limitation, which unfortunately has become quite common in the literature, is to perform analysis with in situ variables (i.e., ClNO2 and N2O5 concentrations) and with variables that will integrate over the air mass's history, such as aerosol nitrate, and then to assume that upwind conditions were similar. This is a major assumption, of course, and many preceding papers spent a lot of time justifying it. It may be useful to add more discussion on what the upwind air masses typically would experience prior to observation (e.g., absence/presence of local sources etc.) at Wangdu.

**Response:** It is true that assuming that upwind conditions were similar is a major assumption. We agree with the reviewer to add a statement on what the upwind air masses typically would experience prior to observation at Wangdu in the selected nights. According to the wind direction and our air masses analysis (Tham et al., 2016), the air mass before arriving at the site was typically influenced by the emission from the nearby villages/cities, coal-fired power plants and biomass burning activities in the region. In our analysis, we carefully select the plumes during the nighttime with certain criteria to make sure a relatively stable period for at least 1.5 hours to perform the analysis as mentioned in the text. For example, we restricted our analysis to data with NO/NO$_x$ ratio lower than 0.1 to remove period with possible

influence from nearby strong $NO_x$ emissions, and the rate of changes for $NO_x/NO_y$ ratio within the period should be smaller than 0.1 $min^{-1}$ to avoid significant changes in the air mass age. To make it clearer, we have added more information and revised the text as follows,

*"The plume age, represented by the ratios of $NO_x$ to $NO_y$, was relatively stable (change <0.1 $min^{-1}$), and no drastic changes were seen in other variables such as the wind conditions, particle surface area, RH, or temperature. Typically, the air masses in the selected cases can be influenced by the emissions from nearby village/urban area, coal-fired power plants and biomass burning activities in the region prior to the arrival at the site (see Tham et al., 2016). Hence, the concentration of NO in the plume must be relatively constant (change of $NO/NO_2$ ratio <0.1 $min^{-1}$) as the presence of a transient NO plume may affect the concentration of $N_2O_5$, which can bias the estimation of $\gamma(N_2O_5)$."*

Reference:

Tham, Y. J., Wang, Z., Li, Q., Yun, H., Wang, W., Wang, X., Xue, L., Lu, K., Ma, N., Bohn, B., Li, X., Kecorius, S., Größ, J., Shao, M., Wiedensohler, A., Zhang, Y., and Wang, T.: Significant concentrations of nitryl chloride sustained in the morning: investigations of the causes and impacts on ozone production in a polluted region of northern China, Atmos. Chem. Phys., 16, 14959-14977, 10.5194/acp-16-14959-2016, 2016.

3) In part because of (2), data were selected in the analysis. While the selection criteria are stated, it is in principle worrisome and may lead to selection bias. Can anything be said about the data that were excluded from analysis? For example, what fraction of the data were excluded, and can you give an indication as to what happens in terms of N2O5 to ClNO2 conversion during those periods - were the mixing ratios of ClNO2 high or low, and was the uptake of N2O5 fast or slow? Could these data be analyzed and added with a lighter shade to some of the Figures?

**Response:** As discussed in major comment (2), we primarily restricted our analysis to data with $NO/NO_x$ ratio lower than 0.1 to remove period with possible influence from nearby strong $NO_x$ emissions, and the rate of changes for $NO_x/NO_y$ ratio within the period should be smaller than 0.1 $min^{-1}$ to avoid significant changes in the air plume age. Sometimes, even if the data comply with the $NO_x$ criteria, we still need to exclude the data based on the mentioned criteria in the text. There are typically two characteristics of the excluded data set if we tried to analyze them. For example, when the data are:
1) 'unstable' conditions within a short period

In the period when the environment condition changes (*e.g.* wind direction and surface area, even though $NO/NO_x$ ratio and the rate of changes for $NO_x/NO_y$ ratio is low (see figure below), the calculated $\gamma(N_2O_5)$ is 0.131, an extremely large value, which we think is not reasonable and we will exclude them from the analysis.

[Figure]

2) 'Non-concurrent' increase or 'bad' correlation for ClNO₂ and NO₃⁻
   As has been stated in the manuscript, the analysis only considered the concurrent increase of both ClNO₂ and NO₃⁻, while the period with decreasing trend in either/both ClNO₂ and/or NO₃⁻ were excluded from the analysis because they will result in negative N₂O₅ uptake and ClNO₂ yield in the calculations.

There are also a few cases that the ClNO₂ and NO₃⁻ increased together but the correlation of the ClNO₂ and NO₃⁻ was weak ($R^2 < 0.5$). Even they may give reasonable values (*e.g.* below $\gamma(N_2O_5) = 0.028$), we will still exclude them due to the high uncertainty from the correlation and may be affected by the changing of the air masses.

[Figure]

Therefore, we think these excluded cases and periods cannot be used to derive the valid uptake coefficient and yield, and thus were not included in the further analysis and the figures. We also revised the text to make it clearer on the fraction of the data was selected, as follows,

*"With these methods and selection criteria, we can derive γ(N₂O₅) and ϕ for 10 different nighttime plumes in 8 out of 13 nights with full CIMS measurement."*

4) The conversion of N2O5 to ClNO2 is often stratified vertically, with usually rapid N2O5 losses at the surface, and higher ClNO2 production rates aloft. How does stratification / vertical mixing affect the analysis?

**Response:** We agree with the reviewer that the production of ClNO$_2$ is closely related to the vertical mixing and the ground-based measurement is always subjected to this phenomenon.

In our previous publication (refer to Tham et al. 2016), we measured a typical nighttime concentration of ClNO$_2$ of about 300 pptv, and the ClNO$_2$ concentration increased up to 2 ppbv after the sunrise. Our model analysis showed that the increase after sunrise is caused by the strong production of ClNO$_2$ in the residual layer during the nighttime and is mixed downward after the break-up of the boundary layer when the sun rises. However, during the nighttime, our previous results suggested that the ClNO$_2$ at Wangdu was mostly produced from the near-surface layer and the mixing between the nocturnal boundary layer and the residual layer is limited. Therefore, we believe that the stratification/vertical-mixing had little or no impact on our analysis of nocturnal N$_2$O$_5$ and ClNO$_2$ at ground level prior to sunrise.

As this issue is still an assumption, we have added the "*limited vertical mixing*" into assumption 1 and a sentence has been added to clarify that this effect is likely not affecting the analysis in the manuscript.

"*Our previous analysis showed that the nighttime vertical mixing is limited at the ground-site of Wangdu (Tham et al., 2016), and likely will not affect the analysis of ClNO$_2$ and NO$_3^-$.*"

Reference:
Tham, Y. J., Wang, Z., Li, Q., Yun, H., Wang, W., Wang, X., Xue, L., Lu, K., Ma, N., Bohn, B., Li, X., Kecorius, S., Größ, J., Shao, M., Wiedensohler, A., Zhang, Y., and Wang, T.: Significant concentrations of nitryl chloride sustained in the morning: investigations of the causes and impacts on ozone production in a polluted region of northern China, Atmos. Chem. Phys., 16, 14959-14977, 10.5194/acp-16-14959-2016, 2016.

**Minor comments:**
page 1 / line 19 - replace "10" with "ten"; state on what basis cases were selected and how the N2O5 uptake coefficients and ClNO2 yields were estimated

**Response:** Revised.

"*The N$_2$O$_5$ uptake coefficient and ClNO$_2$ yield were estimated by using the simultaneously measured ClNO$_2$ and total nitrate in ten selected cases, which have concurrent increases in the ClNO$_2$ and nitrate concentrations and relatively stable environmental conditions.*"

line 21 - grammar: "an average", but then two values (one for N2O5 and one for ClNO2) are given; formatting for the ranges given in brackets is not consistent; The authors should state

their estimated errors of the "observed" N2O5 and ClNO2 uptake parameters here.

**Response:** Thanks for pointing out the issue. The word "an" was removed from the sentence. The ranges have been replaced by the standard deviation values of the observed $N_2O_5$ uptake and $ClNO_2$ yield. The modified sentence in the text is as follows:

"*The determined $\gamma(N_2O_5)$ and $\phi$ values varied greatly, with an average of 0.022 for $\gamma(N_2O_5)$ ($\pm0.012$, standard deviation)) and 0.34 for $\phi$ ($\pm0.28$, standard deviation).*"

line 25 - "by the amount of water in the aerosol, a phenomenon that differs from other field observations". Most models and the Bertram/Thornton parameterization (Eq 3) that contains a water term and would have been included in other field studies. Is the author's statement then really true?

**Response:** Yes, most models and Bertram and Thornton parameterization did include the water and other chemical terms, and yet they cannot reproduce most of the variability of the $N_2O_5$ uptake determined in the field, thus were subjected to a debate on their applicability in 'real' and different environments. Although some laboratory studies (e.g., Thornton et al., 2003; Bertram and Thornton, 2009 and references therein) had found the dependence of $N_2O_5$ uptake on RH and aerosol water content under low water content condition, the field studies previously conducted in Europe and US (e.g., UK, Germany, Boulder, and Texas) did not show a clear dependence of $N_2O_5$ uptake on water content. Previous field studies have linked it with some chemical substances in the aerosol like the nitrate, chloride and organic coatings, as the important factors. Our observation here, however, showed a direct good correlation of $\gamma(N_2O_5)$ with the aerosol water instead of strong dependence on the chemical substances, which was different from other reported field results. We revised the text to make it clearer,

"*...This result suggests that the heterogeneous uptake of $N_2O_5$ in Wangdu is mostly governed by the amount of water in the aerosol, and is strongly water limited, which is different from most of the field observations in the United States and Europe.*"

line 26 - "Laboratory-derived parameterization also overestimated the ClNO2 yield."
Please correct the grammar here.

**Response:** Corrected.

"*The $ClNO_2$ yield estimated from the parameterization was also overestimated comparing to that derived from observation.*"

pg 3/ line 11 - "450" Roberts et al. Geophys. Res. Lett., 36, L20808, 10.1029/2009GL040448, 2009 give a much larger value here; consider adding a second set of Ï¸Tparm calculations with the Roberts et al. value and add to Figure 7a.

**Response:** Roberts et al. (2009) and Behnke et al., (1997) recommended the $k_{R4}/k_{R3}$ values in

the parameterization to be 450 and 836, respectively. We used the value of 483 from Bertram and Thornton (2009), which is in the range of the values from the previous two studies. As Robert's value of 450 is very close to 483 that we used, we then added the results with a value of 836 into our calculation for comparison of parameterized $ClNO_2$ yields, which are also depicted in Figure 7a (figure below) and explained in the figure caption.

[Figure]

*"**Figure 7**: Scatter plots for (a) yield derived from the field versus yield calculated from the parameterization, using $k_{R4}/k_{R3}$ of 483 (recommended by Bertram and Thornton, 2009; solid circle) and 836 (recommended by Behnke et al., 1997; pink open circle). Error bars represent the uncertainty of field-derived $\phi$, the black dotted line represents the 1:1 ratio and the red dotted line shows the quadratic fitting line of the data); (b) field-derived yield versus aerosol water content; (c) field-derived yield versus chloride; and (d) field-derived yield versus $CH_3CN/CO$."*

page 4 / line 11 "We first derive values for $\gamma(N2O5)$ and Ï¸T with the measurement data". Please state briefly here how this is done.

**Response:** A brief statement has been added to the text.

*"We first derive values for $\gamma(N_2O_5)$ and $\phi$ from the regression analysis of $ClNO_2$ and total nitrate ($HNO_3$ and particulate $NO_3^-$) data set and then compare the values obtained in the field with various parameterizations derived from the laboratory studies."*

page 5. Please add a table summarizing the various measurements made. Without one, statements such as "Volatile organic compounds including methane, C2-C10 hydrocarbons, formaldehyde, and oxygenated hydrocarbons and acetonitrile (CH3CN) were measured with a cavity ring–down spectroscopy technique instrument, an on-line gas chromatograph equipped with a mass spectrometer and a ïˇn ´Came ionization detector, a Hantzsch ïˇn ´Cuorimetric monitor, and a proton-transfer-reaction mass spectrometer, respectively" are unnecessarily confusing.

**Response:** Thanks for the suggestion. Actually, a table for instrument list and the measurement techniques, detection limits, time resolution in this campaign has been detailed reported in our previous paper and other publications on this Wangdu campaign. The readers are referred to those publications for more information. We have revised the text to make this clearer,

*"The present study was supported by other auxiliary measurements of aerosol, trace gases, and meteorological parameters, and the detailed instrumentation for the measurement has been listed in a previous paper (Tham et al., 2016)."*

*"Detailed description of these instrumentation and measurement techniques at Wangdu can be found in previous publications (e.g., Wang Y. et al., 2016; Min et al., 2016; and Tham et al., 2016)."*

page 7 / line 23 "what drive" Grammar (either "what drives" or "what factors drive")

**Response:** The phrase has been revised to "*what drives*" as suggested.

page 8 / the "observed" γ(N2O5) is really an aggregate value for N2O5 uptake on the entire aerosol distribution

line 2 - [H2O], [NO3-], and [Cl-] will likely be functions of aerosol size; please add a

disclaimer that this calculation assumes that they are not, and that the predicted gamma values may be biased as a result.

Out of curiosity - is it possible that ClNO2 is produced mainly on sea salt aerosol at Wangdu?

**Response:** Yes, the observed $\gamma(N_2O_5)$ is an aggregate value for uptake on entire aerosol distribution, and the size distribution of different chemical species was not considered in the present study. As stated in the response to major comment (1), a disclaimer has been added in the text as below:

*"It should be noted that the parameterization and calculation here assume an internal mixing of the aerosol chemical species, and the size distribution of [H2O], [NO3-], and [Cl-] in aerosols was not considered due to lack of measurement information. The uptake process would vary with size and mixing state of the particles, thus the predicted γ values here may be biased as a result but represent an average over bulk aerosols."*

For the question on whether the ClNO$_2$ can be produced mainly on sea-salt aerosol at Wangdu, we think that this possibility is low at Wangdu because the site is located about 200 km away from the nearest coast area (Bohai Sea) and the 24 hours air-mass back trajectories showed that there's no indication of marine influence originated air mass. The PM$_{2.5}$ chemical analysis also showed that the chloride to sodium ratio is much higher than the ratio in sea-salt, suggesting that the anthropogenic chloride sources are more important in this location (see Tham et al., 2016 for more information on the chloride source).

line 4 - the E-AIM allows for inclusion of organics, which would alter the liquid water content (maybe). Has this been considered.

**Response:** The effects of the organics on the liquid water content are not considered in this analysis. It is because we have only very limited days/amount of organic data (as stated in the text), which made the analysis harder to consider this organic effect. A quick test-run in the E-AIM Model (IV) was performed by adding the preset organics into the model (*i.e.* malonic acid and succinic acid) with a mixing ratio of $3 \times 10^{-7}$ molar (highest organics level observed in this study period) and 0. The comparison showed that there is only a small difference (<9%) in the liquid water content when considering the organics. We revised the text to be specific on what was used in the model, as follows,

*"The concentration of aerosol liquid water ([H2O]) used in this study was estimated from the E-AIM model IV with inputs of measured bulk aerosol composition of NH4+, Na+, SO4²⁻, NO3⁻ and Cl⁻ (http://www.aim.env.uea.ac.uk /aim/model4/model4a.php) (Wexler and Clegg, 2002), and the V/Sa was taken from the field measurement at Wangdu."*

line 12 what values of Rc and Rp were used in the B&T+org calculation, and are these values realistic for this comparison? (see also major comment 2).

**Response:** The $R_p$ was obtained from the measured median radius of the particle surface area

distribution, with average values about 150 nm. The organic coating thickness L was calculated from the volume ratio of the inorganics to total particles volume following the method in Reimer et al. (2009) by assuming a complete internal mixture. And the $R_c$ was calculated by subtracting the $L$ from $R_p$. We think these values are relatively realistic for this environment as the calculations are based on the measurement of organics in $PM_1$ (from AMS measurement). The assigned density was 1.77 g/cm$^3$ for inorganic density and 1.23 g/cm$^3$ for organics, which are close to the values reported in the measurements of aerosol densities in China (*e.g.* Hu et al., 2012; Li et al., 2016). We have revised the text to make this clearer, as follows,

*"The particle radius $R_p$ was determined from the measured median radius of the particle surface area distribution. The L was calculated from the volume ratio of the inorganics to total particles volume following the method in Reimer et al. (2009) with the assumption of hydrophobic organic coating (density, 1.27 g cm$^{-3}$) on the aqueous inorganic core (with a density of 1.77 g cm$^{-3}$). The aqueous core radius $R_c$ was calculated by subtracting the L from $R_p$."*

pg 9 / line 16 - sulfate should be doubly charged

**Response:** Thanks for pointing out the typo. The sulfate has been changed to double charge.

pg 10 / factors that affect ClNO2 yield - this is an interesting paragraph, but I am a bit skeptical about what appear to be low field yields. Have the authors considered that the lack of agreement may be due to breakdown of the assumptions going into the calculation (uneven distribution of chloride throughout the aerosol, for example)?

**Response:** Yes, we did think of this effect, but we have no information on the size distribution of chloride in the field measurement, and therefore, our conclusion is that the biomass burning activities could partly explain the $ClNO_2$ yield at Wangdu. To clarify, we have revised the statement at the end of the paragraph to acknowledge this possibility and suggest for more future studies of the chloride distribution in the region.

*"The aqueous concentration of Cl⁻ in the present study is relatively higher than previous laboratory studies (e.g., Bertram and Thornton, 2009; Roberts et al., 2009), and might not be fully involved in the reaction R4, for example, the possible effect of nonuniform distribution of chloride within the aerosol. It might contribute to the overestimation and less variability of $\phi$ predicted from the parameterization (Riedel et al., 2013) and the positive relationship of field-derived $\phi$ with $[H_2O]$ (see Figure 7b) might also imply that the increase of water content could increase the availability of the aerosol Cl⁻, thus prompting the reaction R4 to increase the ClNO$_2$ production yield."*

**Response:** In principle yes when the $NO_3^-$ in y-axis and ClNO$_2$ in the x-axis, if we assume that the ClNO$_2$ concentration is zero at sun-set and the air mass does not change in that period.

---

## Author Comment (AC3) · 14 Aug 2018

**General Comments:**
This paper reports an analysis of N2O5 uptake coefficients and ClNO2 yields from a polluted site in the North China Plain during a summer 2014 field intensive. The analysis finds variation of N2O5 uptake coefficients that is characteristic of data sets in other parts of the world. Comparisons between field determinations and laboratory based parameterizations, and between the determined uptake coefficients and other variables, shows that aerosol liquid water / relative humidity is a determining factor. This finding is in contrast to field studies in the U.S. and Europe. ClNO2 yields are shown to be lower than current parameterizations based on the competition between chloride and liquid water, consistent with findings from other regions.
The authors suggest ClNO2 suppression on biomass burning derived particles despite higher chloride content in these aerosol.

Overall, the paper adds to the growing database of these analysis and will be a valuable contribution to the literature. Publication is recommended after the authors address the following comments.

**Response**: We thank the reviewer for his/her attention to this manuscript. We have made all the suggested changes and/or clarifications. The reviewer's comments are in black and our responses are in blue, and the changes in the manuscript are in *italic*.

**Specific comments:**
Page 2, line 4: "yielding N2O5" rather than "yielding a N2O5".

**Response:** Corrected.

Page 3, line 20: Also add Morgan et al., 2015 and McDuffie et al., 2018, to this list.

**Response:** The references have been added to the text (highlighted in yellow).

Page 3, line 22: sentence not clear. Does "laboratory parameterizations can be overestimated" mean that the observations are higher than or lower then the parameterizations?

**Response:** We have revised the sentence to clarify this, as follows:

"*Large discrepancies were observed between the $\gamma(N_2O_5)$ and $\phi$ values determined in the fields and the laboratory parameterizations derived with pure or mixed aerosol samples, and the differences can be up to an order of magnitude.*"

Page 3, line 32: "NO3- aerosol downwind of" rather than "NO3- aerosol in downwind of"

**Response:** The phrase has been revised to *"$NO_3^-$ aerosol downwind of"*.

Page 4, line 8: Is the quoted N2O5 a maximum or an average? Please specify.

**Response:** The $N_2O_5$ level mentioned here is the maximum concentration. It has been revised to *"(1 min-average maximum of 430 pptv)"* in the text.

Page 5, lines 25-26: Are the quoted average production rates of NO3 for nighttime only for nighttime and daytime?

**Response:** It is the average for the night-time only. The word *"night-time"* has been added to the sentence to clarify it.

Page 5, equation 4: The method of Phillips et al. (2016) is referenced, but the method for calculating the production rates in the numerator in the right hand side of the equation is not specified for the data here. How are these quantities (pClNO2 and pNO3-) determined?

**Response:** The $pClNO_2$ and $pNO_3^-$ in the equation 4 were obtained from the slope of linear plot of $ClNO_2$ versus time and $NO_3^-$ versus time, respectively (see the plots below for one example of the selected cases). An additional sentence on determining the pClNO2 and pNO3- have been included in the text, as follows,

*"The $pClNO_2$ and $pNO_3^-$ were determined from the linear fit of the increase of $ClNO_2$ and total $NO_3^-$ (sum of $HNO_3$ and particulate $NO_3^-$) with time, while $[N_2O_5]$ is mean concentration of $N_2O_5$ for the specific duration."*

[Figure]

Example plots for determining the $pClNO_2$ and $pNO_3^-$ used in the case on the night of 29 June.

Page 5, line 33: Define "most nights" – how many nights had r2 > 0.6 for the stated correlation?

**Response:** The sentence has been redefined in the text as the following:

*".... with a coefficient of determination ($r^2$) of greater than 0.6 on 10 out of 13 nights (with*

*full CIMS measurement),"*

Page 6, line 14-15: Assumption 3 is not reasonable. HNO3 is in equilibrium with aerosols regardless of how it is produced. That is, HNO3 equilibration and N2O5 uptake are not separate processes, but tightly coupled ones. The assumption is more likely intended to state that N2O5 heterogeneous uptake during the night of observations is a larger source of total soluble nitrate (HNO3 plus NO3-) than soluble nitrate production from the preceding day, or that the correlation with ClNO2 is determined by the nighttime produced nitrate rather than the background that was present at sunset.

**Response:** Thanks to the reviewer for the suggestion and clarification. We have revised the sentences to make the assumption more reasonable and clearer in the text.

*"...that $N_2O_5$ heterogeneous uptake is a dominant source of total soluble nitrate during the night rather than the gas homogenous production or nitrate production from the preceding daytime."*

Page 6, line 19: remove the word "have"

**Response:** Removed.

Page 6, line 22: Is there a quantitative definition of "drastic changes" here? In other words, is the data filtering arbitrary, or done in a well-defined manner using characteristics of time rates of change.

**Response:** It is difficult to quantify the changes in a well-defined manner for variables like wind direction, RH, temperature and particle surface area for a longer period. These variables will never remain at a constant value in the real environment (*e.g.* the RH is increasing, while the temperature is decreasing with time). These rates of change varied between nights and it is hard to give a 'fix acceptance' values for these changes (*e.g.* it's hard to justify if it has a significant effect for a shift of $20^o$ in the wind direction).

However, for the parameters such as the NO to $NO_x$ ratio and rates of change of $NO_x$ to $NO_y$ ratio, we can filter them in a more defined-manner. For example, we restrict to data with $NO/NO_x$ ratio lower than 0.1 to remove periods with possible influence from nearby strong $NO_x$ emissions, and the rate of changes for $NO_x/NO_y$ ratio within the period should be smaller than 0.1 $min^{-1}$ to avoid significant changes in the air masses.

In other words, we primarily filter the data with the $NO_x$ parameters and then judge and select the period with least changes in other parameters (can be seen in the data in Figure 2a and 2c in the main text) and exclude the data if there's an 'unreasonable' change within the measurement period. We have revised the sentences as below:

*"The plume age, represented by the ratios of $NO_x$ to $NO_y$, were relatively stable (change <*

*0.1 min⁻¹), and no drastic changes were seen in other variables such as the wind conditions,*
*particle surface area, RH, or temperature. Typically, the air masses in the selected cases can*
*be influenced by the emissions from nearby village/urban area, coal-fired power plants and*
*biomass burning activities in the region prior to the arrival at the site (see Tham et al., 2016).*
*Hence the concentration of NO in the plume must be relatively constant (change of NO/NO₂*
*ratio <0.1 min⁻¹), as the presence of a transient NO plume may affect the concentration of*
*N₂O₅, which can bias the estimation of γ(N₂O₅).*"

Reference:
Tham, Y. J., Wang, Z., Li, Q., Yun, H., Wang, W., Wang, X., Xue, L., Lu, K., Ma, N., Bohn, B., Li, X., Kecorius, S., Größ, J., Shao, M., Wiedensohler, A., Zhang, Y., and Wang, T.: Significant concentrations of nitryl chloride sustained in the morning: investigations of the causes and impacts on ozone production in a polluted region of northern China, Atmos. Chem. Phys., 16, 14959-14977, 10.5194/acp-16-14959-2016, 2016.

Page 6, lines 26-28: See comment above. The partitioning of total nitrate between gas and particle phase is an important limitation, and it would be useful to define any quantitative information, such as an aerosol thermodynamic model, that would indicate where this partitioning is. The photochemical soluble nitrate production should be in the background of the correlation (i.e., the intercept) and so might not affect the results.

**Response:** Yes, we agree with the reviewer that the photochemical soluble nitrate production may not affect the results. As for the partitioning, the gas-phase $HNO_3$ measurement showed that it is only 7% (on average) of the total $NO_3^-$ during the nighttime (see Figure 3b in the main text), suggesting that the partitioning from particle to the gas phase is not significant. This information has been included in the text.

"*It is clear that particulate $NO_3^-$ was the dominant species during the night-time at Wangdu, while the nighttime gas-phase $HNO_3$ is only 7% (on average) of the total $NO_3^-$ (Figure 3b).*"

Text on pages 6-7 and Figure 3: Explain why there is significant OH + NO2 during the night. Explain how particulate NO3- production from NO3 + VOC is calculated. Many NO3 + VOC reactions produce organic nitrates rather than HNO3, so it is not clear how this source of HNO3 has been calculated based on the information given. The total production rate of NO3- is also referenced in the text but not shown in the figure. The differentiation between day and night in Figure 3 is not clear. Presumably the time axis is local time, not UTC? Please specify for clarity. The times of day and night should be shown, preferably with a shaded region to indicate night. Data for gas phase HNO3 are presented here for the first time. Why in the preceding analysis was ClNO2 only correlated against particulate phase NO3- if gas phase HNO3 is also available? The analysis should be done from the correlation between ClNO2 and total nitrate (HNO3+ NO3-) since the two are in rapid equilibrium on the time scale of ClNO2 production through N2O5 uptake.

**Response:** Regarding the significant contribution of OH+NO₂ after sunset, it is mostly due to

the non-zero OH concentration, though decreasing towards the night, but is still above the instrument detection limits ($3.2 \times 10^5$ for 30s average, $1\sigma$), together with the significant increase of $NO_2$ level during the night time. Significant levels of OH concentration and reactivity are frequently observed in polluted China environments (*e.g.* Lu et al., 2013; Fuch et al., et al., 2017). The figure below shows the diurnal average of the OH and $NO_2$ for Wangdu during the measurement period.

[Figure]

For the $NO_3$+VOC calculation, we need to clarify that there's a mistake in the figure where the rate of $NO_3$+VOC was already multiplied by 100 times (for it to be 'visible' in the figure), but somehow was not indicated in the legend. As the reviewer suggests, many $NO_3$+VOC reactions, especially the biogenic VOC (e.g. isoprene, alpha-pinene, etc.), produce organic nitrates rather than $HNO_3$. However, there are some $NO_3$+VOC reactions which can produce $HNO_3$ via H abstraction (according to the IUPAC and NIST reaction kinetic datasheet). Some major VOCs, of which measurements are available, have significant concentrations and significant reaction rate, were chosen for the calculation. The table below summarizes the reactions of $NO_3$+VOC used in the analysis. We also need to emphasize that the purpose of this $NO_3$+VOC calculation is just for showing that the $NO_3$+VOC are not a significant source for $HNO_3$ at this site, especially during the nighttime where the $NO_3$ is significant. Therefore, correction on the figure has been made in the text and the information of VOCs used for this calculation has been added in the text too.

| Reaction | Products | $k$ (cm$^3$ molecule$^{-1}$ s$^{-1}$ at 25$^o$ C) |
|---|---|---|
| $NO_3$ + $C_2H_6$ | $\cdot C_2H_5$ + $HNO_3$ | $1.0 \times 10^{-17}$ [a] |
| NO3 + $C_3H_6$ | $\cdot CH_2CH=CH_2$ + $HNO_3$ | $4.8 \times 10^{-16}$ [b] |
| NO3 + $C_3H_8$ | $\cdot CH_3CH_2CH$+ $HNO_3$ $\cdot CH_3CHCH_3$+ $HNO_3$ | $7.0 \times 10^{-17}$ [a] |
| $NO_3$ + HCHO | $\cdot HCO$ + $HNO_3$ | $5.5 \times 10^{-16}$ [a] |
| $NO_3$+ $CH_3OH$ | $\cdot CH_2OH$ + $HNO_3$ | $2.3 \times 10^{-16}$ [b] |
| $NO_3$ + $C_2H_4O$ | $\cdot CH_3CO$ +$HNO_3$ | $2.7 \times 10^{-15}$ [a] |
| $NO_3$+ $CH_3C(O)CH_3$ | $\cdot CH_3C(O)CH_2$ + $HNO_3$ | $3.0 \times 10^{-17}$ [a] |

[a]from IUPAC Atmospheric Chemical Kinetic Data    [b]from NIST Chemical Kinetics Database

The revised text reads,

*"To check the validity of assumptions (2) above, we also calculated the production rate of $NO_3^-/HNO_3$ via reaction of $OH+NO_2$ (=$k_{OH+NO2}[OH][NO_2]$) and $NO_3+VOC$ (=$\Sigma_i k_i[VOC_i][NO_3]$, where $VOC_i$ = $C_2H_6$, $C_3H_6$, $C_3H_8$, HCHO, $CH_3OH$, $C_2H_4O$, $CH_3C(O)CH_3$), as shown in the average diurnal profiles of related species in Figure 3."*

*The revised Figure 3c as below:*

[Figure]

The average $pNO_3^-$ referenced in the text here was determined from the slope of nighttime diurnal particulate $NO_3^-$ in Figure 3b. This information has been added in the text.

The time axis in Figure 3 is local time. This information has been added in the figure caption. Also, an indication of day and night time (shading) has been added in the figure.

The reason that we only correlated the $ClNO_2$ against the particulate phase $NO_3^-$ was that the gas-phase $HNO_3$ concentration during the nighttime was very low (on average about 7% of total $NO_3^-$) and was often below the detection limit (300 pptv) of the measurement by gas and aerosol collector (GAC) (Dong et al., 2012). The inclusion of gas-phase $HNO_3$ in the analysis does not significantly affect the outcome of the $\gamma(N_2O_5)$ and $\phi$, and the changes are still falling within the calculated uncertainty. An example of the difference by adding $HNO_3$ into the analysis can be seen in the figure below. Despite the small changes, we decided to revise all the calculation to include the $HNO_3$ (gas-phase) as suggested by the reviewer to make the analysis more accurate. All the relevant changes have been made in the text.

[Figure]

Example of the difference in the analysis for 20 June case

Reference:

Fuchs, H., et al.: OH reactivity at a rural site (Wangdu) in the North China Plain: contributions from OH reactants and experimental OH budget, Atmos. Chem. Phys., 17, 645-661, 10.5194/acp-17-645-2017, 2017.

Lu, K. D., et al.: Missing OH source in a suburban environment near Beijing: observed and modelled OH and HO2 concentrations in summer 2006, Atmos. Chem. Phys., 13, 1057–1080, doi:10.5194/acp-13-1057-2013, 2013.

Dong, H. B., Zeng, L. M., Hu, M., Wu, Y. S., Zhang, Y. H., Slanina, J., Zheng, M., Wang, Z. F., and Jansen, R.: Technical Note: The application of an improved gas and aerosol collector for ambient air pollutants in China, Atmos. Chem. Phys., 12, 10519-10533, 10.5194/acp-12-10519-2012, 2012.

Page 7, line 22: "The question that arises" rather than "The question arises"

**Response:** The word *"that"* has been added to the sentence.

Page 7, line 33: "coefficients" rather than "coefficient"

**Response:** Corrected.

Page 8, line 21: Figure 5 would be clearer if the field data were on the y-axis and the parameterization on the x-axis.

**Response:** Figure 5 has been revised accordingly.

Page 9, line 22: "to changes in RH" rather than "on the changes in RH"

**Response:** Revised.

Page 9, lines 25-30: Is [H2O]V/Sa really independent of aerosol water itself? It seems that the effects discussed here and on the rest of page 9 can be determined from laboratory

experiments under controlled conditions but not easily determined from field data. The authors should be careful to phrase this argument as consistent with laboratory data rather than a determination of these effects from field measurements.

**Response:** First, we need to clarify that the purpose of correlating the $N_2O_5$ uptake with $[H_2O]V/S_a$ is to show the $\gamma(N_2O_5)$ is increasing with the volume growth (aerosol water and volume of the aerosol).

Second, the major aim of this paragraph on page 9 is to explain the possible reason of good correlation of $\gamma(N_2O_5)$ with aerosol water content, which is consistent with the laboratory experiments, but such an effect has not been seen in other field measurements from the US and Europe and could be an important factor for $N_2O_5$ uptake in China. Therefore, we have rephrased the sentence as below:

*"These results are consistent with several laboratory studies which have demonstrated that an increase in RH enhanced the particle aqueous volume and increased the bulk reactive $N_2O_5$ uptake on aqueous sulfate and organic acids (e.g., malonic, succinic, and glutaric acid) containing aerosols (Thornton et al., 2003; Hallquist et al., 2003)."*

Reference:
Hallquist, M., Stewart, D. J., Stephenson, S. K., and Cox, R. A.: Hydrolysis of N2O5 on sub-micron sulfate aerosols, Phys. Chem. Chem. Phys., 5, 3453-3463, 2003.

Thornton, J. A., Braban, C. F., and Abbatt, J. P.: N2O5 hydrolysis on sub-micron organic aerosols: The effect of relative humidity, particle phase, and particle size, Phys. Chem. Chem. Phys., 5, 4593-4603, 2003.

Page 9, first paragraph: The major conclusion is that RH, and by extension the calculation of aerosol liquid water, was the determining factor for N2O5 uptake. In this context, it will be helpful to say more about the measurement of the wet aerosol surface area and its associated uncertainties, since wet aerosol surface area is often a difficult quantity to measure, and the measurement or calculation can itself introduce an RH dependence to the aerosol surface area measurement. The description in the methods section (Page 5, lines 7-9) is brief. A more comprehensive description of this measurement and statement of its potential dependence on RH, along with the uncertainty in the aerosol surface area, is needed.

**Response**: Thanks for the valuable suggestion. The sentences have been revised and the following information has been added into the text to make it clearer.

*"The particle surface area concentrations ($S_a$) were calculated based on the wet ambient particle number size distribution by assuming spherical particles. In brief, dry-state particle number size distribution was measured with a mobility particle size spectrometer (covering mobility particle diameter of 4 to 800 nm) and an aerodynamic particle size spectrometer (for aerodynamic particle diameter 0.8 to 10 μm). The wet particle number size distributions as a function of the relative humidity were calculated from a size-resolved kappa-Köhler function*

*determined from real-time measurement of a High Humidity Tandem Differential Mobility Analyzer (Hennig et al., 2005; Liu et al., 2014). It should be noted that the major uncertainty of $S_a$ calculation was the assumption and application of $\kappa$ at different size-range, leading to an overall uncertainty of $\pm 19\%$."*

*Reference:*

Hennig, T., Massling, A., Brechtel, F. J., and Wiedensohler, A.: A tandem DMA for highly temperature-stabilized hygroscopic particle growth measurements between 90% and 98% relative humidity, J. Aerosol Sci., 36, 1210-1223, 10.1016/j.jaerosci.2005.01.005, 2005.

Liu, H. J., Zhao, C. S., Nekat, B., Ma, N., Wiedensohler, A., van Pinxteren, D., Spindler, G., Muller, K., and Herrmann, H.: Aerosol hygroscopicity derived from size-segregated chemical composition and its parameterization in the North China Plain, Atmos. Chem. Phys., 14, 2525-2539, 10.5194/acp-14-2525-2014, 2014.

Page 10, line 22, Figure 7a: As for figure 5, this would be clearer with the field data on the y-axis. All other plots in figure 7 have field data on the y-axis, and the same should be done for figure 7.

**Response:** The figures have been edited accordingly.

Page 10, line 24: remove the word "in". Also "Such a discrepancy" rather than "Such discrepancy".

**Response:** The word "in" was removed from the text and the phrase was revised to *"Such a discrepancy".*

Page 10, line 29: What is meant by "from quadratic fitting"? Is there a polynomial fit that should appear in Figure 7?

**Response:** The sentence has been revised to *"correlation from a quadratic fitting"* The Figure7 was edited by adding the quadratic fit line into the plots.

Page 10, line 33: Remove the word "good" or else replace by something more specific, such as "statistically significant", if appropriate. Also, the term "quadratic data fitting" appears again here without explanation or a displayed fit.

**Response:** Thanks for pointing it out. The "good" was removed from the sentence. The display of the quadratic fitting has been added in Figure7.